# StrategyLLM: Large Language Models as Strategy Generators, Executors, Optimizers, and Evaluators for Problem Solving[*]

**Chang Gao**[♣][†]**, Haiyun Jiang**[◇][‡]**, Deng Cai**[♡]**, Shuming Shi**[♡]**, Wai Lam**[♣]
[♣]The Chinese University of Hong Kong  [♡]Tencent AI Lab
[◇]School of Electronic Information and Electrical Engineering, Shanghai Jiao Tong University
{gaochang,wlam}@se.cuhk.edu.hk  haiyunjiangnlp@gmail.com
{jcykcai,shumingshi}@tencent.com

## Abstract

Most existing prompting methods suffer from the issues of generalizability and consistency, as they often rely on instance-specific solutions that may not be applicable to other instances and lack task-level consistency across the selected few-shot examples. To address these limitations, we propose a comprehensive framework, StrategyLLM, allowing LLMs to perform inductive reasoning, deriving general strategies from specific task instances, and deductive reasoning, applying these general strategies to particular task examples, for constructing generalizable and consistent few-shot prompts. It employs four LLM-based agents: strategy generator, executor, optimizer, and evaluator, working together to generate, evaluate, and select promising strategies for a given task. Experimental results demonstrate that StrategyLLM outperforms the competitive baseline CoT-SC that requires human-annotated solutions on 13 datasets across 4 challenging tasks without human involvement, including math reasoning ($34.2\% \rightarrow 38.8\%$), commonsense reasoning ($70.3\% \rightarrow 72.5\%$), algorithmic reasoning ($73.7\% \rightarrow 85.0\%$), and symbolic reasoning ($30.0\% \rightarrow 79.2\%$). Further analysis reveals that StrategyLLM is applicable to various LLMs and demonstrates advantages across numerous scenarios.

## 1   Introduction

Recent advances in large language models (LLMs) have facilitated the development of prompting techniques [26, 43, 20, 8]. In particular, chain-of-thought (CoT) prompting methods [43, 6, 12, 41], which condition LLMs on a few task examples with step-by-step solutions, guide LLMs to break down complex reasoning processes into simpler steps. These approaches have markedly improved performance compared to standard few-shot prompting across a variety of tasks.

Despite their potential, current CoT approaches employing few-shot prompts with instance-specific solutions may face challenges in terms of *generalizability* and *consistency*. Concerning *generalizability*, the solution can be highly specific to the question in each instance, limiting its applicability to other instances. For example, as illustrated in the left part of Figure 1, a solution for a particular system of linear equations with two variables may not provide valuable insights for addressing another system with three variables. Furthermore, the solutions in different instances within the few-shot prompt may exhibit a lack of task-level *consistency*, which complicates the process for LLMs to

---

[*]This research was partially supported by the Center for Perceptual and Interactive Intelligence (CPII) Ltd. under the Innovation and Technology Commission's InnoHK scheme.
[†]Work done during an internship at Tencent AI Lab.
[‡]Haiyun Jiang is the corresponding author.

38th Conference on Neural Information Processing Systems (NeurIPS 2024).

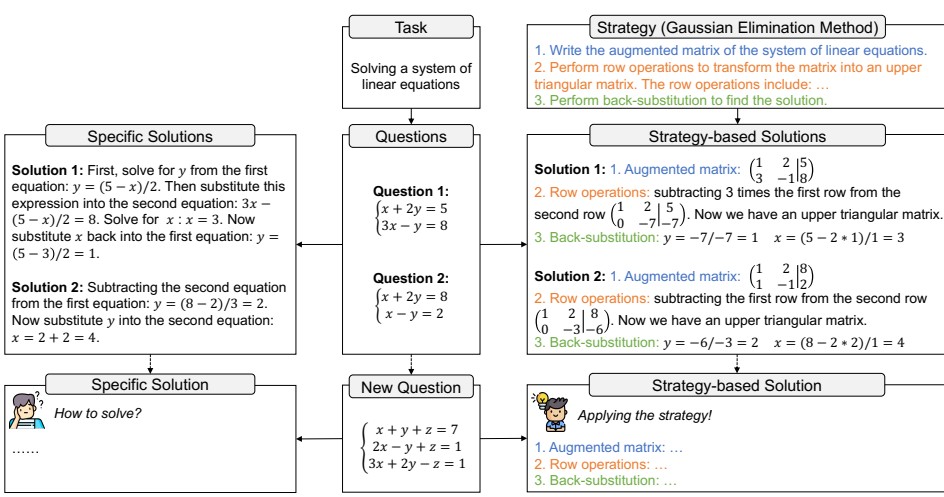

Figure 1: Comparison of specific solutions and strategy-based solutions.

develop effective solutions for tackling new instances. As demonstrated in the left part of Figure 1, the two specific solutions are based on different approaches: Solution 1 employs expression substitution, while Solution 2 utilizes equation subtraction, which may not provide consistent guidance for LLMs to solve new instances. To address these limitations, it is crucial to incorporate *effective problem-solving strategies* and develop *consistent strategy-based solutions* within few-shot prompts. The right part of Figure 1 presents an effective strategy, i.e., Gaussian Elimination Method, offering generalizable steps applicable to any system of linear equations. By providing this strategy and consistently applying it across various instances in the few-shot prompt, LLMs can be better equipped to generate effective solutions for new task instances.

This paper aims to *construct generalizable and consistent strategy-based few-shot prompts for various tasks automatically, while being highly cost-efficient*. Our proposed framework, StrategyLLM, draws inspiration from human cognitive processes to derive general problem-solving strategies. This approach enables LLMs to reason inductively, i.e., deriving general strategies from specific task instances, and deductively, i.e., applying general strategies to particular task examples, to formulate prompts. An example of strategy-based prompts can be seen in Figure 5. The inductive reasoning process enhances *generalizability* by formulating general problem-solving strategies, while the deductive reasoning process improves *consistency* by producing consistent solutions using a given strategy. Developing effective problem-solving strategies is crucial to the success of our framework. To achieve this, we design StrategyLLM as a multi-agent collaboration framework comprising four LLM-based agents—strategy generator, executor, optimizer, and evaluator, as shown in Figure 2. The strategy generator initially creates a pool of strategies that are executed on task examples to assess accuracy, with qualified strategies cached based on a threshold and further evaluated. Unqualified ones may be optimized and re-evaluated iteratively. Through the collaboration of these intelligent agents, our framework is capable of autonomously generating, evaluating, and selecting effective strategies for various tasks and eliminating the need for human involvement.

Crucially, the strategy-based few-shot prompt generation phase is applied once for a given task, after which the learned prompt can be employed for inference on the entire test set. This inference process does not require any additional input beyond the standard few-shot prompting settings. The prompt generation process is highly cost-effective as it necessitates only a few task examples. In particular, StrategyLLM expends less than $0.24 to develop a strategy-based prompt for a variety of tasks using the latest version of GPT-3.5-Turbo.

We conduct comprehensive evaluations of StrategyLLM on 13 datasets across 4 challenging tasks: math reasoning, commonsense reasoning, algorithmic reasoning, and symbolic reasoning. The experimental results reveal the following key findings: (1) StrategyLLM outperforms competitive baselines on all tasks without using any human-annotated reasoning processes; (2) StrategyLLM can be applied to various LLMs and is robust to different groups of task examples; (3) StrategyLLM can generate generalizable and consistent prompts in a cost-effective manner. These findings demonstrate the potential of StrategyLLM as an effective, efficient, and reliable problem-solving framework.

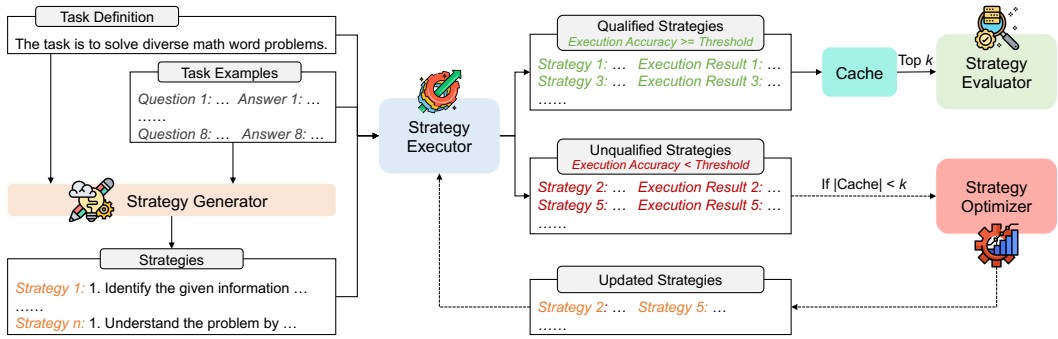

Figure 2: Overview of StrategyLLM. Initially, the strategy generator creates a pool of strategies, which are then applied by the strategy executor to task examples to calculate execution accuracy. Qualified strategies meeting a pre-defined threshold are cached, and if necessary, unqualified strategies are optimized and re-evaluated in iterative cycles. Once a sufficient number of qualified strategies are obtained or the maximum iteration number is reached, the top $k$ strategies are ranked by execution accuracy and evaluated using a validation set.

## 2  StrategyLLM

Our StrategyLLM framework is designed to efficiently create strategy-based few-shot prompts for a wide range of tasks. Subsequently, these prompts can be utilized for inference. In this section, we will introduce our framework in detail. The inference procedure will be discussed in Section 3.

**Overview of StrategyLLM**  As presented in Figure 2, our framework consists of four key agents: strategy generator, executor, optimizer, and evaluator. The prompts for the strategy generator and executor are presented in Figure 3 and Figure 4, respectively. The prompts of the strategy optimizer are in Appendix C. Typically, only a few task examples are used in the collaboration process, making our framework highly efficient.

The collaboration process begins with the strategy generator formulating a pool of strategies based on its understanding of the target task. These strategies then undergo two rounds of validation and selection. In the first round, the strategy executor applies each strategy to a set of task examples to yield its execution result and compute its execution accuracy. Strategies that meet or exceed a pre-set threshold of execution accuracy are deemed qualified and are cached with their corresponding execution results and accuracy. If the number of qualified strategies is less than a pre-defined number $k$, the optimizer refines the unqualified strategies using their execution results. These enhanced strategies are then sent back to the strategy executor for the next iteration. This cycle may repeat until a sufficient number of qualified strategies are achieved or the maximum iteration limit is reached. Following this, all cached strategies are ranked based on their execution accuracy, and the top $k$ strategies are selected. In the second round, the strategy evaluator constructs strategy-based few-shot prompts for each candidate strategy using itself and its execution result and assesses all candidate strategies using their corresponding prompts for inference on a validation set.

**Notations**  We use $p$, $q$, $st$, $so$, and $a$ to denote the prompt, question, strategy, solution, and answer, respectively. During inference, given a question $q$, the language model $M : (p, q) \rightarrow (so, a)$ generates a solution $so$ and an answer $a$ for it conditioned on the prompt $p$. We denote the target task as $t$, its definition as $d$, and the set of task examples as $\mathcal{E}$. Each example in $\mathcal{E}$ is a $(q, a)$ pair.

**Strategy Definition**  In this paper, a task-solving strategy $st$ is defined as a systematic approach designed for universal application across task examples, comprising a series of subtasks that encode task knowledge to address the target task $t$. It is characterized by the following properties: (1) *Task-Level Applicability*: The strategy is formulated in a manner that allows for its application across all task instances, ensuring universality and consistency in its implementation. (2) *Structured Organization*: The strategy comprises a sequence of subtasks that are organized in a logical order to collectively tackle the target task. These subtasks are interconnected and contribute to the overall achievement of the task objective. (3) *Task Knowledge Encoding*: The strategy encapsulates general

Strategy Generator

Task:
{task definition}

Some examples of the task are as follows:
{task examples}

Let's understand the task and write a strategy that consists of a sequence of subtasks to solve the task. For writing, you must satisfy the following requirements:
- Include all necessary subtasks.
- All subtasks are easy to solve.
- Subtasks are in an appropriate order.
- Do not include specific information in the examples.
- Make sure the strategy is general and concise.
The result must be a numbered list in the following format:
1. First subtask
2. Second subtask

Figure 3: Prompt of the strategy generator.

Strategy Executor

Task:
{task definition}

Example of the task:
{example}

Strategy:
{strategy}

The strategy consists of a sequence of subtasks for solving the task. Please execute the strategy on the provided example. For executing, you need to write a step-by-step solution to the example based on the subtasks. The solution must satisfy the following requirements:
- Adjust and execute these subtasks for this example.
- Compute as many intermediate results as possible.
- The answer obtained from the solution must be the same as the original answer.
The result must be in the following format:
Question: Question in the provided example
Solution: Solution obtained based on the subtasks in the strategy
Answer: Answer in the provided example, which must include the string 'The answer is '

Figure 4: Prompt of the strategy executor.

task knowledge and principles, avoiding any specific details unique to individual task examples. These properties collectively contribute to the effectiveness and efficiency of a strategy by promoting consistency, clarity, and informed decision-making in addressing task-level challenges. By embodying these properties, a strategy can serve as a valuable tool to navigate complex tasks and achieve optimal outcomes. An example of the strategy is presented in Figure 5.

**Strategy Generator** $G$    The strategy generator, represented as $G : (\mathcal{E}, d, n) \rightarrow \{st_j\}_{j=1}^n$, aims to generate $n$ diverse strategies for the target task $t$ based on a set of task examples $\mathcal{E}$ and the task definition $d$ using temperature sampling.

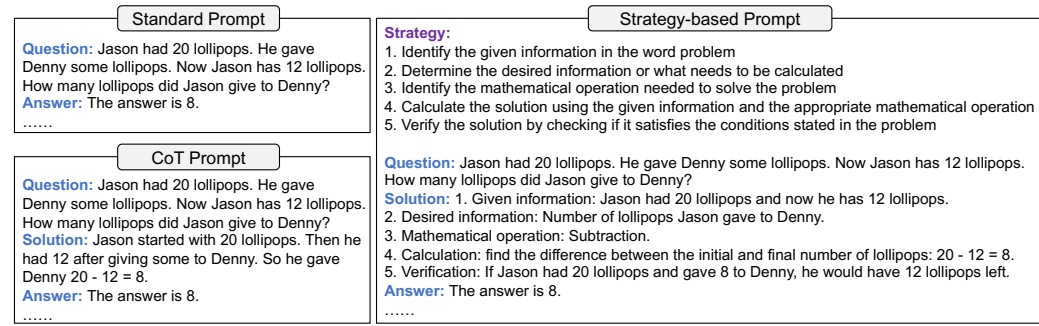

Figure 5: Comparison of the strategy-based, standard, and chain-of-thought (CoT) [43] prompt.

**Strategy Executor** $X$    The strategy executor, denoted as $X : (\mathcal{E}, d, st) \rightarrow (\mathcal{R}_{st}, eacc_{st})$, writes solutions to a set of task examples $\mathcal{E}$ following the strategy $st$ to obtain the execution result $\mathcal{R}_{st} = \{(q, so, a)\}_{i=1}^{|\mathcal{E}|}$ of $st$. The execution accuracy $eacc_{st}$ is calculated as the proportion of examples whose solutions yield correct answers, reflecting the degree of alignment between the strategy and task. Therefore, we select strategies with high execution accuracy as qualified strategies.

**Strategy Optimizer** $O$    The strategy optimizer, represented as $O : (\mathcal{E}, d, st, \mathcal{R}_{st}) \rightarrow st^o$, optimize the strategy $st$ according to its execution result $\mathcal{R}_{st}$ to obtain the updated strategy $st^o$. Firstly, the strategy optimizer $O$ analyzes why some solutions in $\mathcal{R}_{st}$ are not correct and provides suggestions for improving $st$. Secondly, it modifies $st$ to obtain $st^o$ based on the analysis and suggestions.

**Strategy Evaluator** $E$    We select top $k$ candidate strategies according to the execution accuracy. However, to ensure efficiency, we use a limited number of task examples for execution, making the execution accuracy not a very informative metric for choosing strategies. Therefore, we introduce a strategy evaluator to further evaluate the candidate strategies on a validation set $\mathcal{V}$. This process only requires to perform inference once for each candidate strategy and is efficient. The strategy evaluator, denoted as $E : (st, \mathcal{R}_{st}, \mathcal{V}) \rightarrow vacc_{st}$, computes the validation accuracy $vacc_{st}$ of the strategy $st$ on $\mathcal{V}$. To achieve this, it constructs the strategy-based few-shot prompt $p_{st} = (st, \mathcal{R}_{st})$ and conducts inference on $\mathcal{V}$ using $p_{st}$. An example of strategy-based prompts is presented in Figure 5. The validation accuracy $vacc_{st}$ is calculated as the percentage of validation examples whose answers are correct, reflecting the effectiveness of $st$ in real-world scenarios. Strategies with high validation accuracy can be used for inference.

## 3 Inference

Through collaborative efforts among multiple agents, we have obtained multiple candidate strategies, each with its few-shot prompt and validation accuracy. Depending on the task at hand, we can select one or more strategies with high validation accuracy for inference. For simple or specific tasks, a single optimal strategy that solves all task examples effectively may exist, making it sufficient to use only one strategy. However, for complex or diverse tasks, it is unlikely to find a strategy with absolute superiority. In such cases, adopting multiple strategies for inference is more appropriate, as they may excel in different task examples. To harness the strengths of multiple strategies, we employ two methods. The first method involves taking a majority vote on all answers obtained by multiple strategies, akin to the self-consistency (SC) method [41]. The second method requires LLMs to determine the final answer by considering the solutions derived from multiple strategies in a zero-shot (ZS) manner, making it more proper for complex and diverse tasks. We denote the first and second methods as StrategyLLM-SC and StrategyLLM-ZS, respectively. The prompt for the second approach is provided in Appendix D.

## 4 Experiments

### 4.1 Experimental Setup

**Evaluation Tasks and Datasets**    We evaluate StrategyLLM on a variety of tasks:

Table 1: Experimental results on the math reasoning task. The numbers in parentheses represent the relative improvement compared to CoT-SC.

| Methods | AL | PA | IA | CP | NT | GE | PC | Avg |
|---|---|---|---|---|---|---|---|---|
| SP | 32.0 | 50.0 | 17.5 | 27.0 | 20.5 | 21.0 | 20.5 | 26.9 |
| SolutionLLM | 58.5 | 56.5 | 13.5 | 33.0 | 32.0 | 28.0 | 19.5 | 34.4 |
| CoT | 57.0 | 57.5 | 15.0 | 33.5 | 28.0 | 23.0 | 20.0 | 33.4 |
| CoT-SC | 59.0 | 62.0 | 16.5 | 34.5 | 28.0 | 24.5 | 15.0 | 34.2 |
| StrategyLLM | 58.5 | 57.5 | 18.0 | 35.0 | 29.5 | 24.5 | 22.5 | 35.1 |
| StrategyLLM-SC | 60.0 | 61.5 | 18.0 | 38.5 | 30.5 | 28.0 | **24.0** | 37.2 (+8.8%) |
| StrategyLLM-ZS | **64.5** | **65.5** | **19.0** | **39.0** | **32.5** | **28.5** | 22.5 | **38.8 (+13.4%)** |

Table 2: Experimental results on the commonsense, algorithmic, and symbolic reasoning tasks. The numbers in parentheses represent the relative improvement compared to CoT-SC.

| Methods | Commonsense | | | Algorithmic | | | Symbolic | | | |
|---|---|---|---|---|---|---|---|---|---|---|
| | StrategyQA | DU | Avg | WS | MA | Avg | LLC-4 | LLC-8 | LLC-16 | Avg |
| SP | 56.5 | 48.5 | 52.5 | 73.3 | 2.0 | 37.7 | 0 | 0 | 0 | 0 |
| SolutionLLM | 59.5 | 52.0 | 55.8 | 74.7 | 55.3 | 65.0 | 81.5 | 25.5 | 0 | 35.7 |
| CoT | 64.0 | 70.5 | 67.3 | 67.2 | 84.0 | 75.6 | 68.5 | 22.0 | 0 | 30.2 |
| CoT-SC | 70.0 | 70.5 | 70.3 | 61.3 | 86.0 | 73.7 | 68.0 | 22.0 | 0 | 30.0 |
| StrategyLLM | 67.5 | 68.5 | 68.0 | **80.0** | 86.7 | 83.4 | **98.0** | 86.5 | 51.5 | 78.7 |
| StrategyLLM-SC | **71.0** | **74.0** | **72.5 (+3.1%)** | 79.3 | **90.7** | **85.0 (+15.4%)** | **98.0** | **87.0** | **52.5** | **79.2 (+164.0%)** |
| StrategyLLM-ZS | 70.0 | 72.5 | 71.3 (+1.4%) | 78.7 | 89.3 | 84.0 (+14.1%) | **98.0** | 86.0 | 44.0 | 76.0 (+153.3%) |

- **Math Reasoning**: We use the challenging MATH benchmark [16] which comprises problems from mathematics competitions that require more than standard K-12 mathematics tools. It consists of seven datasets of different subjects, namely, Algebra (AL), Prealgebra (PA), Intermediate Algebra (IA), Counting and Probability (CP), Number Theory (NT), Geometry (GE), and Precalculus (PC).

- **Commonsense Reasoning**: We employ StrategyQA [14] and the Date Understanding (DU) task from Big-Bench Hard [38, 9]. StrategyQA necessitates inferring a multi-hop strategy to answer questions, while the DU task involves deducing a date from a given context.

- **Algorithmic Reasoning**: We adopt the Word Sorting (WS) task and the Multi-step Arithmetic (MA) task from Big-Bench Hard [38, 9]. The WS task involves sorting a list of words lexicographically, and the MA task requires solving multi-step equations with basic arithmetic operations.

- **Symbolic Reasoning**: We utilize the Last Letter Concatenation (LLC) task from [43], which requires concatenating the last letters of words in a sequence. In the few-shot prompt, the model only sees examples with two words. To evaluate the generalization abilities of different methods, we construct three out-of-distribution test sets (LLC-4, LLC-8, and LLC-16) with 4, 8, and 16 words in a sequence, respectively.

**Baselines**    We conduct experiments in the few-shot setting and compare StrategyLLM with the following baselines:

- **Standard Prompting (SP)**: SP is the most direct approach for problem-solving. In SP, the prompt $p$ contains a set of question-answer pairs without intermediate reasoning steps.

- **Chain-of-Thought (CoT) Prompting** [43]: CoT incorporates step-by-step solutions for questions in the prompt $p$ to elicit the multi-step reasoning capabilities of LLMs. We use few-shot CoT prompts from [43] for StrategyQA, DU, and LLC, and prompts from [38] for WS and MA. For MATH datasets, we create few-shot CoT prompts by randomly sampling 4 examples from each dataset's training set since these datasets contain human-annotated solutions. The CoT prompts for these datasets are in Appendix H.

- **Self-Consistency with CoT (CoT-SC)** [41]: CoT-SC generates a set of solutions using CoT via temperature sampling to obtain multiple answers. Subsequently, it takes a majority vote over these answers to determine the final answer. For experiments, we sample 3 reasoning paths using temperature sampling with a temperature of 0.7.

Table 3: Experimental results on two math reasoning datasets, namely AL and CP, with different groups of examples.

| Methods | AL-dev | AL-random | CP-dev | CP-random |
|---|---|---|---|---|
| SP | 36.0 | 29.1±3.9 | 25.5 | 26.8±2.5 |
| SolutionLLM | 58.0 | 56.5±2.2 | 31.0 | 32.2±2.8 |
| CoT | 57.5 | 55.1±1.5 | 34.0 | 33.4±1.2 |
| CoT-SC | 59.5 | 58.3±1.2 | 31.5 | 33.0±1.2 |
| StrategyLLM | 57.0 | 54.7±2.5 | 34.5 | 35.6±2.3 |
| StrategyLLM-SC | **64.0** | 58.9±1.1 | 36.5 | 38.4±1.3 |
| StrategyLLM-ZS | 62.5 | **60.8±2.6** | **38.5** | **38.8±1.7** |

- **SolutionLLM**: We construct this baseline to leverage LLMs to directly write the solution for each example in the few-shot prompts using greedy decoding, without using any strategies. The prompt of SolutionLLM is in Appendix E. Since both SolutionLLM and StrategyLLM generate prompts using LLMs, we can eliminate the potential effect of human expertise in the comparison, isolating the impact of incorporating effective strategies.

**Implementation Details**    We employ GPT-3.5 (gpt-3.5-turbo-16k-0613) [32] as the language model for our main experiments, serving as the backend for the strategy generator, executor, optimizer, and evaluator. For a fair comparison with baselines such as CoT, we use the same examples in their few-shot prompts for strategy generation, execution, and optimization. We select the top 1 or 3 strategies with the highest validation accuracy for inference. This allows us to demonstrate the performance of the optimal strategy and the benefits of using multiple strategies. We adopt greedy decoding for inference. Details of the strategies for each dataset can be found in Appendix G. The validation set size is 100 for all the datasets. For datasets with over 200 test examples, we randomly sample 200 examples for testing to reduce API costs. More details can be found in Appendix B.

## 4.2   Main Results

Tables 1 and 2 present the experimental results of StrategyLLM and several baselines across four reasoning tasks. We have the following observations:

- *StrategyLLM is an effective and efficient framework for problem-solving*. StrategyLLM using multiple strategies, i.e., StrategyLLM-SC and StrategyLLM-ZS, outperforms all baselines across the four reasoning tasks. Furthermore, StrategyLLM employing the best discovered strategy consistently outperforms CoT and SolutionLLM. Notably, StrategyLLM automatically constructs generalizable and consistent few-shot prompts for tackling various tasks without human expertise, while CoT relies on human-annotated examples for each task.

- *Explicitly incorporating effective strategies significantly enhance the complex reasoning and out-of-distribution (OOD) generalization abilities of LLMs*. For example, our framework demonstrates more considerable improvements on the MATH benchmark compared to the simpler commonsense reasoning datasets. Furthermore, StrategyLLM substantially surpasses CoT and SolutionLLM on the three OOD test sets of the LLC task, showcasing the generalizability of effective strategies.

- *Adopting multiple strategies brings obvious benefits on complex or diverse tasks*. The performance of StrategyLLM is significantly improved by using multiple strategies on the math and commonsense reasoning tasks. The benefits of leveraging multiple strategies on simpler or more specific tasks, i.e., symbolic and algorithmic reasoning, is less significant. These observations indicate that our framework is capable of creating multiple complementary strategies for diverse or complex tasks. Furthermore, StrategyLLM-ZS outperforms StrategyLLM-SC on the math reasoning task, showing that allowing LLMs to determine the answer is more appropriate for intricate tasks.

## 5   Analysis

**Evaluating the robustness of StrategyLLM**    We conduct an investigation to assess the robustness of our StrategyLLM framework with respect to varying groups of examples. For this purpose, we

Table 4: Experimental results of closed-source models on the CP, StrategyQA, and MA datasets. The numbers in parentheses represent the relative improvement compared to CoT-SC.

| Methods | GPT-4 | | | | Claude-3-Sonnet | | | |
|---|---|---|---|---|---|---|---|---|
| | CP | StrategyQA | MA | Avg | CP | StrategyQA | MA | Avg |
| SolutionLLM | 52.0 | 75.5 | 96.7 | 74.7 | 21.0 | 73.5 | 69.3 | 54.6 |
| CoT | 49.5 | 80.5 | 92.7 | 74.2 | 26.0 | 69.0 | 72.7 | 55.9 |
| CoT-SC | 54.5 | **83.5** | 94.7 | 77.6 | 26.0 | 75.0 | 76.7 | 59.2 |
| StrategyLLM | 52.5 | 81.5 | **98.7** | 77.2 | **28.0** | 75.0 | 83.3 | 62.1 |
| StrategyLLM-SC | **56.0** | **83.5** | **98.7** | **79.4** (+2.4%) | **28.0** | **77.0** | **88.0** | **64.3** (+8.6%) |

Table 5: Experimental results of open-source models on the CP, StrategyQA, and MA datasets. The numbers in parentheses represent the relative improvement compared to CoT-SC.

| Methods | Meta-Llama-3-8B-Instruct | | | | Meta-Llama-3-70B-Instruct | | | |
|---|---|---|---|---|---|---|---|---|
| | CP | StrategyQA | MA | Avg | CP | StrategyQA | MA | Avg |
| SolutionLLM | 20.5 | 64.0 | 43.3 | 42.6 | 51.5 | 79.0 | 72.0 | 67.5 |
| CoT | 16.0 | 61.0 | 44.7 | 40.6 | 48.5 | 80.5 | 81.3 | 70.1 |
| CoT-SC | 19.5 | 71.0 | 45.3 | 45.3 | 47.0 | 81.5 | 82.0 | 70.2 |
| StrategyLLM | 24.5 | **74.0** | 64.7 | 54.4 | 51.5 | 82.0 | 88.0 | 73.8 |
| StrategyLLM-SC | **25.0** | **74.0** | **66.0** | **55.0** (+21.5%) | **54.0** | **83.5** | **91.3** | **76.3** (+8.7%) |

| Methods | Mixtral-8x7B-Instruct-v0.1 | | | | Mixtral-8x22B-Instruct-v0.1 | | | |
|---|---|---|---|---|---|---|---|---|
| | CP | StrategyQA | MA | Avg | CP | StrategyQA | MA | Avg |
| SolutionLLM | 22.5 | 61.0 | 34.7 | 39.4 | 44.5 | 72.0 | 60.7 | 59.1 |
| CoT | 24.5 | 63.0 | 59.3 | 48.9 | 41.0 | 72.0 | 80.0 | 64.3 |
| CoT-SC | 26.5 | 73.5 | 62.7 | 54.2 | 40.5 | 75.0 | 80.7 | 65.4 |
| StrategyLLM | 28.5 | 73.5 | 76.0 | 59.3 | 44.5 | 76.5 | 84.0 | 68.3 |
| StrategyLLM-SC | **32.0** | **75.0** | **78.0** | **61.7** (+13.7%) | **47.5** | **77.0** | **89.3** | **71.3** (+9.0%) |

select two math reasoning datasets with diverse examples, namely AL and CP, and randomly sample 5 distinct groups of examples from their respective training sets. We then report the mean and standard deviation of the results. Additionally, we employ the validation set to identify a group of 4 examples from the training set. Specifically, we use the OpenAI embedding model API (the text-embedding-3-large model) to map training and validation questions to embeddings and subsequently select the 4 training examples with the highest cosine similarities to all validation examples. We designate these groups of examples as AL-dev and CP-dev, respectively. The results, as presented in Table 3, demonstrate that StrategyLLM consistently delivers satisfactory performance on both datasets, suggesting that StrategyLLM is a robust and reliable framework for problem-solving.

**Exploring the universality of StrategyLLM** To investigate the universality of our StrategyLLM framework, we apply it to a variety of LLMs to evaluate its effectiveness. For closed-source models, we utilize GPT-4 (gpt-4-0613) [31] and Claude-3-Sonnet (claude-3-sonnet-20240229) [2]. For open-source models, we employ Meta-Llama-3-8B-Instruct, Meta-Llama-3-70B-Instruct [1], Mixtral-8x7B-Instruct-v0.1, and Mixtral-8x22B-Instruct-v0.1 [18]. We conduct experiments on the CP, StrategyQA, and MA datasets, which represent three distinct reasoning tasks. The results, summarized in Tables 4 and 5, reveal that integrating effective strategies for constructing generalizable and consistent few-shot prompts yields significant benefits across a range of model capabilities and task complexities, underscoring the framework's universality. StrategyLLM notably enhances performance in open-source models such as Meta-Llama-3-8B-Instruct and Mixtral-8x7B-Instruct-v0.1, particularly on the CP and MA datasets which demand complex reasoning, indicating the effectiveness of our framework in scenarios requiring sophisticated problem-solving. These findings further corroborate StrategyLLM's robustness and reliability as a problem-solving framework.

**Comparing reasoning via task-level strategy and instance-specific planning** Our framework facilitates generalizable and consistent reasoning by developing task-level strategies. To evaluate

Table 6: Comparison of Plan-and-Solve, CoT+Strategy, and StrategyLLM.

| Methods | CP | StrategyQA | MA | Avg |
|---------|------|------|------|------|
| Plan-and-Solve | 26.0 | 54.0 | 69.3 | 49.8 |
| Plan-and-Solve-SC | 27.5 | 64.5 | 70.0 | 54.0 |
| CoT+Strategy | 30.5 | 63.0 | 62.7 | 52.1 |
| CoT+Strategy-SC | 36.5 | 70.0 | 70.0 | 58.8 |
| StrategyLLM | 35.0 | 67.5 | 86.7 | 63.1 |
| StrategyLLM-SC | **38.5** | **71.0** | **90.7** | **66.7** |

the necessity of effective task-level strategies, we compare our framework against two baselines: (1) Plan-and-Solve Prompting [40], which directs LLMs to formulate specific plans for each test instance at inference time and execute these plans to solve the instances; (2) CoT+Strategy, which combines the CoT prompt with instructions that guide LLMs to devise a task-solving strategy and apply it to a specific test example at inference time. The prompt for CoT+Strategy is detailed in Appendix F.

The performance of GPT-3.5 on the CP, StrategyQA, and MA datasets, representing three distinct reasoning tasks, is presented in Table 6. Our observations are as follows: (1) StrategyLLM significantly outperforms both Plan-and-Solve Prompting and CoT+Strategy across all three datasets. This highlights the superiority of generalizable task-level strategies over instance-specific plans in enhancing performance across various problem-solving contexts. This improvement can be attributed to two key factors: (a) our task-level strategies encapsulate essential task-level knowledge, thereby providing professional and high-level guidance; (b) generating high-quality, specific plans for each test example at inference time is inherently challenging, making it difficult to ensure the quality of these plans. (2) Even when explicitly encouraged to devise a general task-solving strategy in the CoT+Strategy method, the LLM tends to produce strategies that are highly specific to the test example and encode limited task-level knowledge. This underscores the necessity of creating generalizable strategy-based few-shot prompts.

**Analyzing the cost of strategy-based prompt generation**  In this analysis, we evaluate the cost of the strategy-based prompt generation process. The process includes the strategy generator, executor, optimizer, and evaluator, each contributing to the overall cost for each reasoning task. Table 7 details the average cost incurred by our StrategyLLM framework in generating a candidate strategy-based prompt, calculated by dividing the total cost of the process by the number of candidate strategies $k$. The costs are presented in terms of input and output tokens and the money associated with using GPT-3.5-Turbo. The results indicate that our framework is economically efficient. The average cost for `gpt-3.5-turbo-16k-0613` ranges from \$0.33 to \$1.12 across the four reasoning tasks. For the latest version of GPT-3.5-Turbo, specifically `gpt-3.5-turbo-0125`, the cost is considerably lower, ranging from \$0.08 to \$0.24. Generally, tasks of higher complexity consume more tokens due to their inherently longer solutions.

**Examining results across various difficulty levels**  The problems in the MATH benchmark are classified by difficulty on a scale of 1 to 5. The easiest problems are assigned a difficulty level of 1, while the most challenging problems are given a difficulty level of 5. Figure 6 illustrates the performance of CoT-SC and StrategyLLM-SC on the seven datasets within the MATH benchmark, considering different difficulty levels. It is evident that the enhanced performance of StrategyLLM-SC over CoT-SC stems from its ability to tackle more complex problems, underscoring the significance of generalizable strategies in augmenting intricate reasoning.

More analysis can be found in Appendix A.

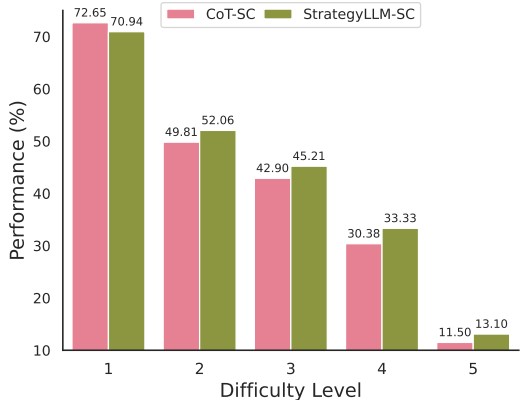

Figure 6: Comparison of CoT-SC and StrategyLLM-SC performance on the MATH benchmark across various difficulty levels.

Table 7: Average cost of prompt generation across four reasoning tasks.

|  | Math | Commonsense | Algorithmic | Symbolic |
|---|---|---|---|---|
| # Input Tokens | 287.83K | 228.67K | 107.27K | 70.94K |
| # Output Tokens | 63.14K | 33.15K | 32.95K | 28.48K |
| Cost of `gpt-3.5-turbo-16k-0613` | $1.12 | $0.82 | $0.45 | $0.33 |
| Cost of `gpt-3.5-turbo-0125` | $0.24 | $0.16 | $0.10 | $0.08 |

# 6 Related Work

**Prompting LLMs for Problem Solving**   The prominent chain-of-thought (CoT) prompting approach [43] has inspired a variety of prompting methods aimed at enhancing the problem-solving abilities of LLMs. These methods include using programming languages to describe the reasoning process [6, 13, 28], representing the reasoning process with complex structures such as trees or graphs [46, 3, 36, 47], applying task decomposition [49, 19, 34, 4], implementing self-correction with automatic feedback [22, 29, 30, 5, 7], and combining different prompting techniques [27, 50]. However, most of these approaches require manual annotation of reasoning processes, limiting their generalizability and flexibility. By comparison, our StrategyLLM framework can automatically construct strategy-based few-shot prompts for any task, ensuring generalizable and consistent solutions following effective strategies. This approach sets it apart from existing automatic prompt construction methods [48, 37, 45], which may generate inconsistent solutions within the prompt. The plan-and-solve prompting method [40] aims to address missing-step errors by requesting LLMs to generate a plan before solving a specific example in a zero-shot manner. The plan is instance-specific and significantly different from the task-solving strategy which can be applied to all task examples. The learning-to-plan approach [15] learns a text plan for each task to assist LLMs in problem-solving. The plan, which is not necessarily a strategy, can be any instruction helpful for solving the task. Moreover, it demands a large training and validation set during the learning process, resulting in high costs. In contrast, our framework is efficient and cost-effective.

**LLM-based Autonomous Agents**   The adoption of autonomous agents driven by LLMs across various disciplines is revolutionizing our methodologies for tackling problems, making decisions, and fostering innovation [39, 44]. These agents have been utilized to enhance the reasoning capabilities of LLMs [42, 24, 11], contribute to social simulation [33, 23, 25, 21], and advance software development [35, 17, 10]. In this paper, we employ multiple LLM-based agents to collaborate in the generation, execution, optimization, and evaluation of problem-solving strategies.

# 7 Discussion

**Limitation and Impact**   The key idea behind StrategyLLM is to harness the knowledge and reasoning capabilities of LLMs to develop and refine task-solving strategies tailored to specific tasks. By utilizing the extensive knowledge embedded in these LLMs, which are trained on diverse data sources spanning multiple domains, StrategyLLM is able to generate generalizable strategies that incorporate domain-specific expertise. However, if the model possesses limited knowledge in a particular domain, it is unlikely to create effective strategies for that domain. In such cases, merely optimizing the prompt may not significantly improve performance, and domain-specific continual training may be necessary. As LLMs continue to expand their knowledge bases and enhance their reasoning capabilities, their ability to generate generalizable strategies for diverse tasks is expected to improve, implying the potential of our StrategyLLM framework.

**Conclusion**   This paper proposes StrategyLLM, harnessing the power of LLMs to construct generalizable and consistent few-shot prompts for various tasks efficiently. Our framework's effectiveness and reliability are substantiated through extensive evaluations on four challenging tasks: mathematical reasoning, commonsense reasoning, algorithmic reasoning, and symbolic reasoning. Further analysis reveals that our framework exhibits robustness across different task example groups, application to various LLMs, cost-efficiency in prompt generation, and effectiveness in complex reasoning.

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

# A  Additional Analysis

**Examining the optimization process**    In our primary experiments, the strategy generator initially produces 30 strategies, and we aim to obtain 10 qualified strategies for validation. For complex datasets, it is difficult to directly acquire 10 qualified strategies without optimization. Table 8 presents the optimization process for three challenging datasets: CP from the mathematical reasoning task, StrategyQA from the commonsense reasoning task, and MA from the algorithmic reasoning task, along with the iteration in which the optimal strategy (i.e., the strategy with the highest validation accuracy) is obtained. It is evident that the strategy optimizer plays a vital role in obtaining more qualified strategies and superior strategies, indicating its significance in our framework.

Table 8: Total count of qualified strategies achieved in each iteration of the optimization process and the specific iteration when the optimal strategy is attained.

| Iteration | CP | StrategyQA | MA |
|---|---|---|---|
| 1 | 7 | 9 | 6 |
| 2 | 12 | 10 | 9 |
| 3 | - | - | 10 |
| **Optimal** | 2 | 1 | 2 |

**Analyzing the inference cost**    In this section, we assess the inference cost associated with the optimal strategy-based prompt generated by our StrategyLLM framework, as well as the inference costs of the SolutionLLM and CoT baselines. The costs are represented by the average input and output tokens required for each test example, which are displayed in Table 9. Generally, StrategyLLM consumes more tokens during inference compared to CoT and SolutionLLM, as its prompt encompasses both general strategies and step-by-step solutions adhering to these strategies. In this paper, our primary objective is to develop a framework capable of generating generalizable and consistent prompts for various tasks without human intervention. To reduce inference costs, we may encourage LLMs to create more succinct strategies by imposing additional constraints during the prompt generation process, which will be explored in our future work.

In our main experiments, we utilize 4 examples in the few-shot prompts of datasets within the MATH benchmark. To compare StrategyLLM with baselines of comparable costs, we construct CoT baselines comprising 8 few-shot examples, with average input and output tokens on the datasets of the MATH benchmark amounting to 2697 and 288, respectively. This baseline is referred as CoT-8, and its first 4 examples are the same as CoT in Table 1. The comparison results between CoT-8 and StrategyLLM are presented in Table 10. Our framework achieves a marked improvement over CoT-8, showcasing its effectiveness.

**Examining the impact of prompt consistency**    In our primary experiments, we have demonstrated that StrategyLLM outperforms inconsistent CoT prompts. To further examine the impact of prompt consistency, we intentionally create inconsistent prompts by employing multiple strategies. For each test example within a specific dataset, we generate an inconsistent few-shot prompt for it by randomly selecting examples from different strategy-based prompts. Specifically, we apply the top 3 strategies randomly and uniformly to the examples in the prompt. As a result, the prompt fails to

Table 9: Average inference cost of each test example across four reasoning tasks. # I and # O denote the number of input and output tokens, respectively.

| | Math | Commonsense | Algorithmic | Symbolic |
|---|---|---|---|---|
| SolutionLLM (# I) | 1693 | 740 | 373 | 301 |
| SolutionLLM (# O) | 373 | 90 | 103 | 48 |
| CoT (# I) | 1332 | 487 | 830 | 261 |
| CoT (# O) | 304 | 50 | 331 | 65 |
| StrategyLLM (# I) | 2649 | 2139 | 888 | 842 |
| StrategyLLM (# O) | 511 | 279 | 227 | 244 |

Table 10: Experimental results on the math reasoning task. The numbers in parentheses represent the relative improvement compared to CoT-8-SC.

| Methods | AL | PA | IA | CP | NT | GE | PC | Avg |
|---|---|---|---|---|---|---|---|---|
| CoT-8 | 56.5 | 59.5 | **20.5** | 28.5 | 27.0 | 21.0 | 22.0 | 33.6 |
| CoT-8-SC | 61.0 | 57.5 | 17.0 | 38.5 | 29.0 | 23.5 | 18.0 | 34.9 |
| StrategyLLM | 58.5 | 57.5 | 18.0 | 35.0 | 29.5 | 24.5 | 22.5 | 35.1 |
| StrategyLLM-SC | 60.0 | 61.5 | 18.0 | 38.5 | 30.5 | 28.0 | **24.0** | 37.2 (+6.5%) |
| StrategyLLM-ZS | **64.5** | **65.5** | 19.0 | **39.0** | **32.5** | 28.5 | 22.5 | **38.8 (+11.1%)** |

Table 11: Results of StrategyLLM using the best discovered strategy and the method employing inconsistent prompts.

| Methods | CP | StrategyQA | MA | Avg |
|---|---|---|---|---|
| Inconsistent Prompt | 29.0 | 56.5 | 77.3 | 54.3 |
| StrategyLLM | **35.0** | **67.5** | **86.7** | **63.1** |

offer consistent guidance for LLMs, requiring them to choose the most appropriate strategy for each test example based on their understanding of the strategies and the test example itself. We evaluate the effect of prompt consistency on the CP, StrategyQA, and MA datasets. Table 11 presents the comparison between this method and StrategyLLM using the best discovered strategy. It is evident that the approach employing inconsistent prompts performs considerably worse than StrategyLLM, indicating that automatically identifying the most suitable strategy for each test example is quite difficult. Therefore, consistently applying an effective strategy to various examples within the prompt is advantageous.

**Assessing the complementarity of strategies**   To this end, we employ multiple strategies to derive various solutions and ascertain the answer by majority voting. Specifically, we employ the top 1, 3, 5, 7, and 9 strategies for the CP, StrategyQA, and MA datasets. Figure 7 unveils the following observations: (1) StrategyLLM-SC consistently surpasses CoT-SC on all three datasets when employing multiple solutions, suggesting that explicitly introducing effective strategies to obtain solutions is beneficial; (2) Leveraging multiple strategies outperforms the utilization of a single strategy across all three datasets, implying that multiple complementary strategies exist in most scenarios; (3) Incorporating additional strategies generally demonstrates advantageous. However, this does not guarantee enhancement in performance, as it relies on the effectiveness of the newly introduced strategies and their complementarity with pre-existing strategies.

**Upper limit of accuracy with multiple strategies**   The upper limit of utilizing multiple strategies can be determined by calculating the *coverage*, which is defined as the percentage of examples that can be accurately solved by at least one strategy. The coverage represents the maximum potential accuracy achievable with multiple strategies. Figure 8 illustrates the coverage and accuracy (i.e.,

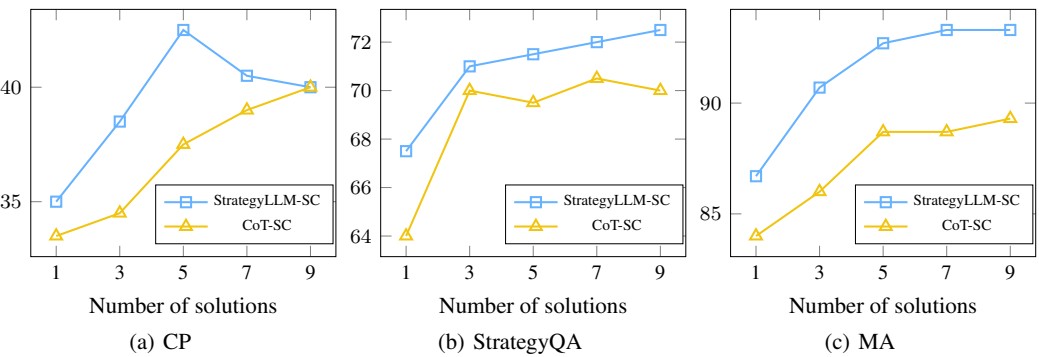

(a) CP  (b) StrategyQA  (c) MA

Figure 7: Performance of StrategyLLM-SC and CoT-SC on the CP, StrategyQA, and MA datasets.

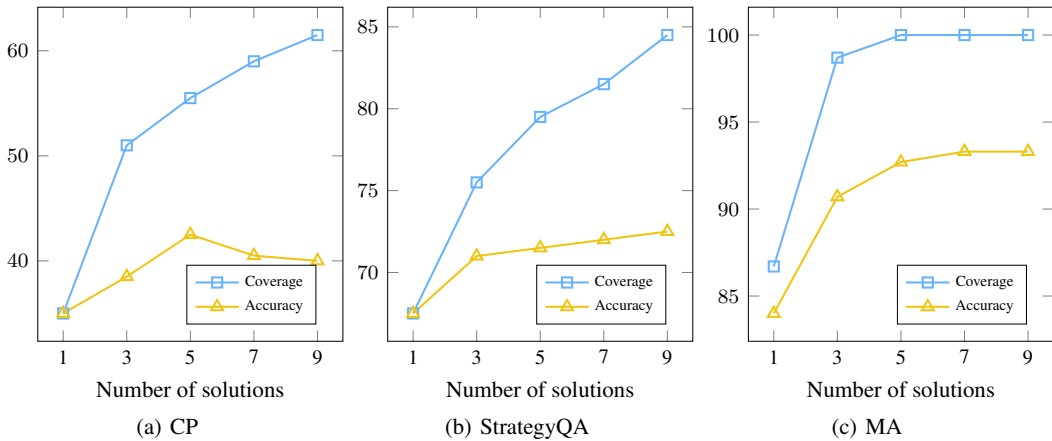

|     |     |     |
| :-: | :-: | :-: |
| (a) CP | (b) StrategyQA | (c) MA |

Figure 8: Coverage and accuracy of StrategyLLM using multiple strategies on the CP, StrategyQA, and MA datasets.

through a majority vote) of the StrategyLLM using multiple strategies on the CP, StrategyQA, and MA datasets. We can observe that there is a significant gap between accuracy and coverage, indicating that substantial performance improvements could be realized if the optimal strategy for each test example is selected rather than relying on a simple majority vote.

**Case study**  In this paragraph, we present a case study to demonstrate the advantages of StrategyLLM over CoT reasoning. We utilize an example with a difficulty level of 5 from the CP dataset, as illustrated in Figure 9, which includes the example and solutions provided by both StrategyLLM and CoT. While CoT offers a flexible reasoning process, it can sometimes lead to inconsistencies or omissions in understanding and solving problems. Specifically, in the provided example, CoT mishandles rotations and reflections and fails to provide adequate details on how to address them. In contrast, StrategyLLM is designed to reduce the likelihood of misinterpretations and enhance reasoning through a structured approach. The generalizable strategy employed by StrategyLLM facilitates a comprehensive understanding of the problem, effective utilization of relevant concepts, and a systematic breakdown of the task. This structured approach ensures that critical details are not overlooked, thereby leading to the correct solution. This case study clearly highlights the superiority of StrategyLLM in handling complex reasoning tasks. Beyond accuracy, the solutions derived from StrategyLLM are more detailed, interpretable, and accessible, making them particularly suitable for contexts that demand clarity and comprehensibility, such as educational settings.

## B   Implementation Details

The strategy generator creates $n = 30$ strategies using temperature sampling with a temperature of 1. The strategy executor, optimizer, and evaluator employ greedy decoding with a temperature of 0. The threshold of the execution accuracy for all datasets is set to 0.75, except for the CP and GE datasets. Their thresholds are set to 0.5 due to the challenging examples in their few-shot prompts. The maximum iteration number for strategy optimization is 3. We retain $k = 10$ candidate strategies for validation, selecting the top 1 or 3 strategies with the highest validation accuracy for inference.

## C   Prompts of Strategy Optimizer

For optimizing a strategy, the strategy optimizer conducts the following tasks: (1) constructing the examination result for each task example by comparing the answer extracted from its execution result and the gold answer, which is demonstrated in Figure 10; (2) analyzing the examination results of all provided task examples and providing suggestions for improving the strategy using the prompt in Figure 11; (3) modifying the strategy to obtain an updated strategy based on the feedback, i.e., the analysis and suggestions, leveraging the prompt in Figure 12.

## D   Prompt of StrategyLLM-ZS

The prompt is shown in Figure 13.

## E   Prompt of SolutionLLM

The prompt is presented in Figure 14.

## F   Prompt of CoT+Strategy

The prompt is demonstrated in Figure 15.

## G   Strategies and Execution Results

Figures 16-27 provide the best strategies discovered for various datasets and their execution results on task examples obtained by GPT-3.5. Figures 28-30 present the best strategies discovered for various datasets and their execution results on task examples obtained by GPT-4.

## H   Chain-of-Thought Prompts for MATH Datasets

Figures 31-39 demonstrate the chain-of-thought prompts for the seven datasets from the MATH benchmark.

**Question:** How many ways are there to arrange 6 beads of distinct colors in a $2 \times 3$ grid if reflections and rotations are considered the same? (In other words, two arrangements are considered the same if I can rotate and/or reflect one arrangement to get the other.)
**Gold Answer:** 180

**CoT Solution:**
We can consider the $2 \times 3$ grid as a 6-element cycle. There are 6! ways to arrange the 6 beads in a line, but we must divide by 6 to account for rotations. However, this overcounts the number of arrangements by a factor of 2 because we are also counting reflections. Therefore, the final answer is $\frac{6!}{6 \cdot 2} = 60$.
**Answer:** The answer is 60.

**StrategyLLM Solution:**
1. Understand the problem statement:
- We have 6 beads of distinct colors.
- We need to arrange these beads in a 2x3 grid.
- Reflections and rotations are considered the same, meaning two arrangements are considered the same if one can be obtained from the other by rotating or reflecting.

2. Identify the counting or probability concept:
- We need to use the concept of combinations to solve this problem.

3. Break down the problem:
- We have 6 beads to arrange in a 2x3 grid.
- We need to consider the symmetries of the grid to eliminate redundant cases.
- We need to determine the number of distinct arrangements considering reflections and rotations.

4. Calculate the required counting or probability:
- To calculate the number of distinct arrangements, we need to count the number of combinations that satisfy the given conditions.
- We can use combinations to calculate this.

- Let's consider the possible arrangements:
- We can choose any of the 6 beads for the first position in the grid.
- We can choose any of the remaining 5 beads for the second position.
- We can choose any of the remaining 4 beads for the third position.
- We can choose any of the remaining 3 beads for the fourth position.
- We can choose any of the remaining 2 beads for the fifth position.
- We can choose the last remaining bead for the sixth position. - Total number of distinct arrangements = 6 x 5 x 4 x 3 x 2 x 1 = 720.

5. Simplify or reduce the problem:
- We need to consider the symmetries of the grid to eliminate redundant cases.
- The 2x3 grid has rotational symmetry of order 2, meaning it can be rotated 180 degrees to look the same.
- The 2x3 grid also has reflectional symmetry, meaning it can be reflected horizontally or vertically to look the same.
- Therefore, we need to divide the total number of distinct arrangements by 2 to account for the rotational symmetry.
- We also need to divide the total number of distinct arrangements by 2 to account for the reflectional symmetry.

6. Calculate the total number of distinct arrangements:
- Total number of distinct arrangements = 720 / 2 / 2 = 180.

**Answer:** The answer is 180.

Figure 9: Case study.

> **Strategy Optimizer - Construction of Examination Result**
>
> Example:
> {example}
>
> Execution result obtained by executing the strategy on the example:
> {execution result}
>
> The answer extracted from the execution result is {predicted answer}, and the correct answer is {gold answer}. They are {different or the same}.

Figure 10: Construction of the examination result for a task example.

> **Strategy Optimizer - Analysis & Suggestion**
>
> Task:
> {task definition}
>
> Strategy:
> {strategy}
>
> Examination results obtained by executing the strategy on the provided examples of the task and examining the execution results:
> {examination results}
>
> We can see that we do not get the correct answer after executing this strategy on some of the provided examples. Please carefully analyze why the answers extracted from the execution results of these examples are incorrect and provide suggestions for improving the strategy.

Figure 11: Analysis & Suggestion Prompt of the strategy optimizer.

> **Strategy Optimizer - Modification**
>
> Task:
> {task definition}
>
> Some examples of the task are as follows:
> {task examples}
>
> Original strategy to solve the task:
> {original strategy}
>
> Feedback:
> {feedback}
>
> You need to modify the original strategy based on the feedback to obtain a better strategy. The newly obtained strategy must be a numbered list in the following format:
> 1. First subtask
> 2. Second subtask

Figure 12: Modification Prompt of the strategy optimizer.

StrategyLLM-ZS

Question:
{question}

Candidate Solutions:
{candidate solutions}

Please determine the final answer for the question based on the candidate solutions and their corresponding answers. The output should be in the following format:
Final Answer: The final answer, which must include the string 'The final answer is '.

Figure 13: Prompt of StrategyLLM-ZS.

SolutionLLM

Task:
{task definition}

Example of the task:
{example}

Please write a solution to the provided example. The answer obtained from the solution must be the same as the original answer. The result must be in the following format:
Question: Question in the provided example
Solution: Solution to the question
Answer: Answer in the provided example, which must include the string 'The answer is '

Figure 14: Prompt of SolutionLLM.

CoT+Strategy

{CoT Prompt}

We show you several task examples of how to generate solutions. Please develop a general strategy that consists of a sequence of subtasks to solve the task and apply the strategy to the following question to obtain its solution and answer.
Question: {question}
The result must be in the following format:
Strategy: Strategy employed to solve the task
Question: Provided question
Solution: Solution obtained by applying the strategy to the question
Answer: Answer to the question, which must include the string 'The answer is '

Figure 15: Prompt of CoT+Strategy.

**Strategy**:
1. Simplify expressions involving radicals by multiplying and combining under a single radical if possible.
2. Rationalize the denominators of fractions with radicals by multiplying the numerator and denominator by the conjugate of the denominator.
3. Use the quadratic formula or factoring to find the roots of quadratic equations.
4. Find the coordinates of points of intersection between two curves by solving the system of equations formed by setting the two equations equal to each other.
5. Find the product or sum of the roots of a quadratic equation by applying Vieta's formulas.

**Executing the strategy on a task example**:
**Question:** The quadratic $8x^2 + 12x - 14$ has two real roots. What is the sum of the squares of these roots? Express your answer as a common fraction in lowest terms.
**Solution:**
1. Simplify expressions involving radicals by multiplying and combining under a single radical if possible.
- There are no radicals in the given quadratic equation, so no simplification is needed.
2. Rationalize the denominators of fractions with radicals by multiplying the numerator and denominator by the conjugate of the denominator.
- There are no fractions with radicals in the given quadratic equation, so no rationalization is needed.
3. Use the quadratic formula or factoring to find the roots of quadratic equations.
- The quadratic formula is given by: $x = \frac{-b \pm \sqrt{b^2 - 4ac}}{2a}$, where $ax^2 + bx + c = 0$.
- For the given quadratic equation $8x^2 + 12x - 14$, we have $a = 8$, $b = 12$, and $c = -14$.
- Plugging these values into the quadratic formula, we get:
$x = \frac{-12 \pm \sqrt{12^2 - 4(8)(-14)}}{2(8)}$
$x = \frac{-12 \pm \sqrt{144 + 448}}{16}$
$x = \frac{-12 \pm \sqrt{592}}{16}$
$x = \frac{-12 \pm 4\sqrt{37}}{16}$
$x = \frac{-3 \pm \sqrt{37}}{4}$
- Therefore, the two real roots of the quadratic equation are $\frac{-3 + \sqrt{37}}{4}$ and $\frac{-3 - \sqrt{37}}{4}$.
4. Find the sum of the squares of the roots.
- Let $r_1$ and $r_2$ be the roots of the quadratic equation.
- The sum of the squares of the roots is given by $r_1^2 + r_2^2$.
- Plugging in the values of the roots, we get:
$r_1^2 + r_2^2 = \left(\frac{-3 + \sqrt{37}}{4}\right)^2 + \left(\frac{-3 - \sqrt{37}}{4}\right)^2$
$r_1^2 + r_2^2 = \frac{(-3 + \sqrt{37})^2}{4^2} + \frac{(-3 - \sqrt{37})^2}{4^2}$
$r_1^2 + r_2^2 = \frac{9 - 6\sqrt{37} + 37}{16} + \frac{9 + 6\sqrt{37} + 37}{16}$
$r_1^2 + r_2^2 = \frac{2(9 + 37)}{16}$
$r_1^2 + r_2^2 = \frac{2(46)}{16}$
$r_1^2 + r_2^2 = \frac{92}{16}$
$r_1^2 + r_2^2 = \frac{23}{4}$
**Answer:** The answer is $\frac{23}{4}$.

Figure 16: Strategy generated by GPT-3.5 for MATH-Algebra and its execution on a task example.

## MATH-Prealgebra

**Strategy**:
1. Understand the problem: Read and carefully analyze the given problem to understand the requirements and constraints.
2. Identify the known and unknown information: Determine what information is provided and what needs to be calculated or determined.
3. Translate the problem into mathematical equations or statements: Use appropriate mathematical symbols, equations, or statements to represent the problem.
4. Solve the equations or statements: Use algebraic techniques to solve the mathematical equations or statements.
5. Verify the Solution: Substitute the found values back into the original problem to check if they satisfy the given conditions.
6. Express the solution in the required format: Answer the question or provide the solution in the appropriate form, such as a fraction, decimal, or percentage.

**Executing the strategy on a task example**:
**Question:** Pete thinks of a number. He doubles it, adds 10, multiplies by 4, and ends up with 120. What was his original number?
**Solution:**
1. Understand the problem: Pete starts with a number, performs a series of operations on it, and ends up with a final result. We need to determine the original number.
2. Identify the known and unknown information:
Known information: Pete ends up with 120 after doubling the number, adding 10, and multiplying by 4.
Unknown information: The original number Pete started with.
3. Translate the problem into mathematical equations or statements:
Let x be the original number Pete started with.
Pete doubles the number: $2x$
Pete adds 10: $2x + 10$
Pete multiplies by 4: $4(2x + 10)$
Pete ends up with 120: $4(2x + 10) = 120$
4. Solve the equations or statements:
$4(2x + 10) = 120$
Distribute the 4: $8x + 40 = 120$
Subtract 40 from both sides: $8x = 80$
Divide both sides by 8: $x = 10$
The original number Pete started with is 10.
5. Verify the Solution: Substitute the found value back into the original problem.
Pete doubles 10: $2(10) = 20$
Pete adds 10: $20 + 10 = 30$
Pete multiplies by 4: $4(30) = 120$
The final result is indeed 120, which matches the given information.
6. Express the solution in the required format:
**Answer:** The answer is 10.

Figure 17: Strategy generated by GPT-3.5 for MATH-Prealgebra and its execution on a task example.

## MATH-Intermediate Algebra

**Strategy**:
1. Begin by identifying the type of problem you are given: quadratic polynomial, algebraic equation, geometric problem, etc.
2. If the problem involves a polynomial, start by determining the degree of the polynomial and then proceed to solve for the unknown coefficients using the given conditions or given equations.
3. If the problem involves algebraic equations, simplify the expressions and solve for the unknowns by isolating the variable. Make use of algebraic rules such as distributive property, solving equations with fractions, and solving systems of equations if necessary.
4. If the problem involves geometric concepts, analyze the given information and understand the geometric relationships involved. Apply the appropriate formulas or theorems and solve for the unknown quantities.
5. If the problem involves inequalities, solve for the range of possible values by analyzing the conditions or constraints given. Make use of algebraic manipulation and reasoning skills to determine the range or the maximum/minimum values.
6. If the problem involves functions, analyze the properties of the function and use any given information to determine the unknown variables, such as finding the maximum/minimum values, solving for x-intercepts or y-intercepts, or finding asymptotes.
7. Check your solutions by substituting them back into the original equation or problem statement to ensure they satisfy the given conditions.

**Executing the strategy on a task example**:
**Question:** Find the quadratic polynomial $p(x)$ such that $p(-3) = 10$, $p(0) = 1$, and $p(2) = 5..$
**Solution:**
1. Identify the type of problem: quadratic polynomial.
2. Determine the degree of the polynomial and solve for the unknown coefficients using the given conditions.
Let the quadratic polynomial be $p(x) = ax^2 + bx + c$.
Using the given conditions:
$p(-3) = 10 \Rightarrow a(-3)^2 + b(-3) + c = 10$
$p(0) = 1 \Rightarrow a(0)^2 + b(0) + c = 1$
$p(2) = 5 \Rightarrow a(2)^2 + b(2) + c = 5$
Simplifying the equations:
$9a - 3b + c = 10$
$c = 1$
$4a + 2b + c = 5$
Substituting $c = 1$ into the equations:
$9a - 3b + 1 = 10$
$4a + 2b + 1 = 5$
Simplifying further:
$9a - 3b = 9$
$4a + 2b = 4$
3. Solve the system of equations using algebraic manipulation.
Multiplying the second equation by 3:
$9a - 3b = 9$
$12a + 6b = 12$
Adding the equations together: $21a = 21$
Dividing both sides by 21: $a = 1$
Substituting $a = 1$ into the second equation:
$4(1) + 2b = 4$
$4 + 2b = 4$
$2b = 0$
Dividing both sides by 2: $b = 0$
Substituting $a = 1$ and $b = 0$ into the first equation:
$9(1) - 3(0) = 9$
$9 = 9$
Therefore, $a = 1$, $b = 0$, and $c = 1$.
4. Write the quadratic polynomial $p(x)$.
$p(x) = 1x^2 + 0x + 1$
Simplifying:
$p(x) = x^2 + 1$
**Answer:** The answer is $x^2 + 1$.

Figure 18: Strategy generated by GPT-3.5 for MATH-Intermediate Algebra and its execution on a task example.

---

**MATH-Counting and Probability**

**Strategy**:
1. Understand the problem statement:
- Read and understand the given problem.
- Identify what needs to be calculated or determined.
- Make note of any special conditions or restrictions mentioned.
2. Identify the counting or probability concept:
- Determine which counting or probability concept is needed to solve the problem.
- Common concepts include permutations, combinations, probability rules, etc.
3. Break down the problem:
- Break down the problem into smaller subproblems or cases if needed.
- Identify any patterns or symmetries that can simplify the calculations.
- Consider any specific conditions or restrictions mentioned in the problem.
4. Calculate the required counting or probability:
- Apply the appropriate counting or probability formula or rule to calculate the desired quantity.
- Use combinations or permutations for counting problems.
- Apply probability rules like the multiplication rule, addition rule, conditional probability, etc., for probability problems.
5. Simplify or reduce the problem:
- Look for ways to simplify or reduce the problem.
- Utilize symmetry or equivalence to eliminate redundant cases.
- Consider any combinatorial identities or properties that can simplify calculations.
6. Verify and check the Solution:
- Double-check the calculations to ensure accuracy.
- Check if the solution satisfies any specific conditions or restrictions mentioned in the problem.
- Compare the solution with any provided answers or results.
7. Communicate the solution clearly:
- Present the final solution in a clear and concise manner.
- Include any necessary explanations or steps taken to arrive at the solution.
- Express the final answer in the required units or as a fraction, decimal, or whole number, as specified by the problem.
8. Practice and review:
- Solve additional problems on counting and probability to reinforce the concepts.
- Review the solved problems to identify any common patterns or strategies that can be applied to similar problems.

**Executing the strategy on a task example**:
**Question:** Given that $\binom{17}{7} = 19448$, $\binom{17}{8} = 24310$ and $\binom{17}{9} = 24310$, calculate $\binom{19}{9}$.
**Solution:**
1. Understand the problem statement:
- We are given three values of binomial coefficients: $\binom{17}{7} = 19448$, $\binom{17}{8} = 24310$, and $\binom{17}{9} = 24310$.
- We need to calculate $\binom{19}{9}$.
2. Identify the counting or probability concept:
- We need to use the concept of combinations to solve this problem.
3. Break down the problem:
- We can use the identity $\binom{n}{k} = \binom{n-1}{k-1} + \binom{n-1}{k}$ to break down the problem.
- Using this identity, we can express $\binom{19}{9}$ in terms of the given values: $\binom{19}{9} = \binom{18}{8} + \binom{18}{9}$.
4. Calculate the required counting or probability:
- Using the given values, we can calculate $\binom{18}{8}$ and $\binom{18}{9}$.
- $\binom{18}{8} = \binom{17}{7} + \binom{17}{8} = 19448 + 24310 = 43758$
- $\binom{18}{9} = \binom{17}{8} + \binom{17}{9} = 24310 + 24310 = 48620$
5. Simplify or reduce the problem:
- We have calculated the values of $\binom{18}{8}$ and $\binom{18}{9}$.
- We can now substitute these values into the expression for $\binom{19}{9}$: $\binom{19}{9} = \binom{18}{8} + \binom{18}{9} = 43758 + 48620 = 92378$
6. Verify and check the Solution:
- The solution obtained is 92378, which matches the provided answer.
7. Communicate the solution clearly:
- The value of $\binom{19}{9}$ is 92378.
**Answer:** The answer is 92378.

---

Figure 19: Strategy generated by GPT-3.5 for MATH-Counting and Probability and its execution on a task example.

## MATH-Number Theory

**Strategy**:
1. Understand the problem: Read the problem carefully and identify what is being asked and what information is given.
2. Apply basic number theory concepts: Use basic number theory principles such as modular arithmetic, divisibility rules, prime factorization, etc., to simplify the problem or find relevant information.
3. Simplify expressions: Simplify any given expressions or equations to make them easier to work with.
4. Solve congruences: If the problem involves solving congruences (e.g., finding the remainder when a number is divided by another number), use the Chinese Remainder Theorem or specific methods for solving modular equations to find the solution.
5. Study patterns and properties: Look for patterns or properties in the given problem that might help in finding the solution. This could involve using modular arithmetic properties, Euler's totient function, or other relevant principles.
6. Use prime factorization: If the problem involves finding factors or multiples of numbers, use the prime factorization of the numbers involved to simplify the calculations.
7. Apply number theory theorems: If the problem relates to specific number theory theorems, such as Fermat's Little Theorem or Wilson's Theorem, apply those theorems to find the solution.
8. Check for divisibility: Use divisibility rules to check if a number is divisible by another number, or find other relevant information that can help in solving the problem.
9. Solve step-by-step: Break down the problem into smaller steps and solve them one by one to find the final solution.
10. Check the Answer: Always double-check your final answer and ensure that it satisfies the given conditions or requirements of the problem.

**Executing the strategy on a task example**:
**Question:** What is the remainder when 2007 is divided by 81?
**Solution:**
1. Understand the problem: We are asked to find the remainder when 2007 is divided by 81.
2. Apply basic number theory concepts: We can use the concept of modular arithmetic to solve this problem. In modular arithmetic, we find the remainder when a number is divided by another number.
3. Simplify expressions: There are no expressions to simplify in this problem.
4. Solve congruences: To find the remainder when 2007 is divided by 81, we can write it as a congruence: $2007 \equiv x \pmod{81}$, where $x$ is the remainder we want to find.
5. Study patterns and properties: There are no specific patterns or properties to consider in this problem.
6. Use prime factorization: We can use the prime factorization of 81 to simplify the calculations. $81 = 3^4$.
7. Apply number theory theorems: There are no specific number theory theorems to apply in this problem.
8. Check for divisibility: We can check if 2007 is divisible by 81. If it is divisible, then the remainder will be 0. Otherwise, we need to find the remainder.
9. Solve step-by-step:
a. Divide 2007 by 81: $2007 \div 81 = 24$ with a remainder of 63.
b. The remainder is 63.
10. Check the Answer: The remainder we found, 63, is the same as the original answer.
**Answer:** The answer is 63.

Figure 20: Strategy generated by GPT-3.5 for MATH-Number Theory and its execution on a task example.

**Strategy**:
1. To find the area of a shaded region in a circle sector overlap problem:
a. Calculate the area of each sector using the formula $A_{\text{sector}} = \frac{1}{2}r^2\theta$, where $r$ is the radius and $\theta$ is the central angle.
b. Determine the area of the overlapping region by subtracting the area of the triangle formed by the two radii from the sum of the areas of the two sectors.
c. Double-check the calculations and verify the obtained answer against the correct answer.
2. To calculate the perimeter of a quadrilateral formed by cutting a square with a line:
a. Find the coordinates of the intersection points between the line and the square.
b. Calculate the lengths of the sides of the quadrilateral using the distance formula.
c. Add up the lengths of the sides to find the perimeter.
d. Double-check the calculations and verify the obtained answer against the correct answer.
3. To determine the height of a cylinder inscribed in a hemisphere:
a. Use the Pythagorean theorem to find the radius of the cylinder's base by subtracting the height of the cylinder from the radius of the hemisphere.
b. The height of the cylinder is equal to the radius of the hemisphere minus the radius of the cylinder's base.
c. Double-check the calculations and verify the obtained answer against the correct answer.
4. To reflect a parallelogram across the x-axis:
a. Change the sign of the y-coordinate of each vertex to obtain the reflected image of the parallelogram.
b. Double-check the coordinates of the reflected vertices and verify against the correct answer.
5. To reflect a parallelogram across a given line:
a. Find the equation of the given line.
b. Calculate the distance between each vertex of the parallelogram and the given line.
c. Reflect each vertex across the given line by moving it the same distance on the opposite side of the line.
d. Connect the reflected vertices to form the reflected parallelogram.
e. Double-check the coordinates of the reflected vertices and verify against the correct answer.
6. To find the ordered pair of a reflected point:
a. Determine the equation of the line of reflection.
b. Use the formula for the reflection of a point across a line to find the coordinates of the reflected point.
c. Double-check the coordinates of the reflected point and verify against the correct answer.
7. Double-check all calculations and results to ensure accuracy.
a. Verify the obtained answers against the correct answers.
b. Review all steps and calculations to identify any errors or discrepancies.
c. Make necessary corrections and adjustments to improve the accuracy of the solution.

**Executing the strategy on a task example**:
**Question:** Parallelogram $ABCD$ with $A(2,5)$, $B(4,9)$, $C(6,5)$, and $D(4,1)$ is reflected across the $x$-axis to $A'B'C'D'$ and then $A'B'C'D'$ is reflected across the line $y = x + 1$ to $A''B''C''D''$. This is done such that $D'$ is the image of $D$, and $D''$ is the image of $D'$. What is the ordered pair of $D''$ in the coordinate plane?
**Solution:**
1. Reflect parallelogram $ABCD$ across the x-axis to obtain parallelogram $A'B'C'D'$: a. Change the sign of the y-coordinate of each vertex: - $A'(2,-5)$ - $B'(4,-9)$ - $C'(6,-5)$ - $D'(4,-1)$
2. Reflect parallelogram $A'B'C'D'$ across the line $y = x + 1$ to obtain parallelogram $A''B''C''D''$:
a. Find the equation of the line $y = x + 1$: - The slope of the line is 1, and the y-intercept is 1. - The equation of the line is $y = x + 1$.
b. Calculate the distance between each vertex of parallelogram $A'B'C'D'$ and the line $y = x + 1$: - Distance from $A'$ to the line $y = x + 1$: - Substitute the coordinates of $A'$ into the equation of the line: - $-5 = 2 + 1$ - $-5 = 3$ - The distance is $|-5-3| = 8$.
- Distance from $B'$ to the line $y = x + 1$: - Substitute the coordinates of $B'$ into the equation of the line: - $-9 = 4 + 1$ - $-9 = 5$ - The distance is $|-9-5| = 14$.
- Distance from $C'$ to the line $y = x + 1$: - Substitute the coordinates of $C'$ into the equation of the line: - $-5 = 6 + 1$ - $-5 = 7$ - The distance is $|-5-7| = 12$.
- Distance from $D'$ to the line $y = x + 1$: - Substitute the coordinates of $D'$ into the equation of the line: - $-1 = 4 + 1$ - $-1 = 5$ - The distance is $|-1-5| = 6$.
c. Reflect each vertex of parallelogram $A'B'C'D'$ across the line $y = x + 1$ by moving it the same distance on the opposite side of the line: - Reflect $A'$ across the line $y = x + 1$: - Move $A'$ 8 units in the opposite direction of the line's normal vector: - $A''(2 - 8, -5 + 8) = (-6, 3)$
- Reflect $B'$ across the line $y = x + 1$: - Move $B'$ 14 units in the opposite direction of the line's normal vector: - $B''(4 - 14, -9 + 14) = (-10, 5)$
- Reflect $C'$ across the line $y = x + 1$: - Move $C'$ 12 units in the opposite direction of the line's normal vector: - $C''(6 - 12, -5 + 12) = (-6, 7)$
- Reflect $D'$ across the line $y = x + 1$: - Move $D'$ 6 units in the opposite direction of the line's normal vector: - $D''(4 - 6, -1 + 6) = (-2, 5)$                27
3. The ordered pair of $D''$ is $(-2, 5)$.
**Answer:** The answer is $(-2, 5)$.

**Strategy**:
1. Identify the problem or concept being tested in the given question.
2. Define any necessary terms or formulas related to the problem.
3. Simplify or manipulate any given expressions or equations to solve the problem.
4. Solve the problem step by step with the help of appropriate formulas or techniques.
5. Verify the solution or answer to ensure its correctness.
6. Repeat steps 1-5 for each problem in the precalculus subject.

**Executing the strategy on a task example**:
**Question:** The function

$$f(z) = \frac{(-1 + i\sqrt{3})z + (-2\sqrt{3} - 18i)}{2}$$

represents a rotation around some complex number $c$. Find $c$.
**Solution:**
1. Identify the problem or concept being tested in the given question.
- The problem is asking us to find the complex number $c$ that represents the center of rotation for the given function.
2. Define any necessary terms or formulas related to the problem.
- The formula for a rotation around a complex number $c$ is given by $f(z) = e^{i\theta}(z - c) + c$, where $\theta$ is the angle of rotation.
3. Simplify or manipulate any given expressions or equations to solve the problem.
- The given function can be rewritten as $f(z) = \frac{-z + i\sqrt{3}z - 2\sqrt{3} - 18i}{2}$.
4. Solve the problem step by step with the help of appropriate formulas or techniques.
- Comparing the given function with the formula for rotation, we can see that $e^{i\theta} = \frac{-1 + i\sqrt{3}}{2}$ and $c = \frac{-2\sqrt{3} - 18i}{2}$. - To find $\theta$, we can take the argument of $e^{i\theta}$, which is $\theta = \arg\left(\frac{-1 + i\sqrt{3}}{2}\right)$.
- Simplifying the expression inside the argument, we get $\frac{-1 + i\sqrt{3}}{2} = \frac{-2 + 2i\sqrt{3}}{4} = \frac{-1}{2} + \frac{i\sqrt{3}}{2}$.
- The argument of this complex number is $\theta = \arctan\left(\frac{\sqrt{3}}{-1}\right) = \arctan(-\sqrt{3})$. - Using the unit circle, we can determine that $\theta = -\frac{\pi}{3}$. - Therefore, the angle of rotation is $-\frac{\pi}{3}$.
5. Verify the solution or answer to ensure its correctness.
- Substituting $\theta = -\frac{\pi}{3}$ into the formula for rotation, we get $f(z) = e^{-i\frac{\pi}{3}}(z - c) + c$. - Simplifying this expression, we have $f(z) = \frac{1}{2}\left(\cos\left(-\frac{\pi}{3}\right) + i\sin\left(-\frac{\pi}{3}\right)\right)(z - c) + c$. - Using the trigonometric identities $\cos(-\frac{\pi}{3}) = \frac{1}{2}$ and $\sin(-\frac{\pi}{3}) = -\frac{\sqrt{3}}{2}$, we can further simplify the expression to $f(z) = \frac{1}{2}\left(\frac{1}{2} - \frac{\sqrt{3}}{2}i\right)(z - c) + c$. - Distributing and simplifying, we get $f(z) = \frac{1}{4}(z - c) - \frac{\sqrt{3}}{4}i(z - c) + c$. - Combining like terms, we have $f(z) = \left(\frac{1}{4} - \frac{\sqrt{3}}{4}i\right)z + \left(\frac{3}{4} + \frac{\sqrt{3}}{4}i\right)c$. - Comparing this expression with the given function, we can see that $\frac{1}{4} - \frac{\sqrt{3}}{4}i = \frac{-1 + i\sqrt{3}}{2}$ and $\frac{3}{4} + \frac{\sqrt{3}}{4}i = \frac{-2\sqrt{3} - 18i}{2}$. - Therefore, our solution is correct.
**Answer:** The answer is $\sqrt{3} - 5i$.

Figure 22: Strategy generated by GPT-3.5 for MATH-Precalculus and its execution on a task example.

**StrategyQA**

**Strategy**:
1. Read and understand the question fully by carefully analyzing the given information and requirements.
2. Identify the type of question being asked, whether it is a simple "yes or no" question or if there is a need to provide a specific answer.
3. Determine the key elements and concepts mentioned in the question that need to be addressed.
4. Break down the question into smaller parts, if necessary, to identify any hidden information or assumptions.
5. Look for any clues or hints within the question that can help in finding the answer.
6. Decide on the approach to be used, whether it requires logical reasoning, knowledge of specific facts, comparison, or analysis.
7. Use prior knowledge and general understanding to answer the question, if applicable.
8. If additional information is required, either use external sources or gather relevant data from memory.
9. Organize the information and answer the question clearly and concisely.
10. Review the answer to ensure it accurately addresses the question and provides a correct response.
11. If necessary, double-check the answer to ensure it aligns with the requirements and request.
12. Finally, present the answer in a clear and understandable manner, either as a "yes or no" response or as a specific answer as required by the question.

**Executing the strategy on a task example**:
**Question:** Could Brooke Shields succeed at University of Pennsylvania?
**Solution:**
1. Read and understand the question fully by carefully analyzing the given information and requirements.
- The question is asking if Brooke Shields could succeed at the University of Pennsylvania.
2. Identify the type of question being asked, whether it is a simple "yes or no" question or if there is a need to provide a specific answer.
- The question requires a simple "yes or no" answer.
3. Determine the key elements and concepts mentioned in the question that need to be addressed.
- Key elements: Brooke Shields, University of Pennsylvania, succeed.
4. Break down the question into smaller parts, if necessary, to identify any hidden information or assumptions.
- No need to break down the question further.
5. Look for any clues or hints within the question that can help in finding the answer.
- The question does not provide any specific clues or hints.
6. Decide on the approach to be used, whether it requires logical reasoning, knowledge of specific facts, comparison, or analysis.
- The approach requires logical reasoning and knowledge of the University of Pennsylvania.
7. Use prior knowledge and general understanding to answer the question, if applicable.
- The University of Pennsylvania is a prestigious university known for its rigorous academic programs. Brooke Shields, as a successful actress and model, may have the potential to succeed at the university.
8. If additional information is required, either use external sources or gather relevant data from memory.
- No additional information is required.
9. Organize the information and answer the question clearly and concisely.
- The answer is yes, Brooke Shields could succeed at the University of Pennsylvania.
10. Review the answer to ensure it accurately addresses the question and provides a correct response.
- The answer accurately addresses the question and provides a correct response.
11. If necessary, double-check the answer to ensure it aligns with the requirements and request.
- The answer aligns with the requirements and request.
12. Finally, present the answer in a clear and understandable manner, either as a "yes or no" response or as a specific answer as required by the question.
**Answer:** The answer is yes.

Figure 23: Strategy generated by GPT-3.5 for StrategyQA and its execution on a task example.

**Date Understanding**

**Strategy**:
1. Determine the current date based on the given information.
2. Calculate the number of hours or days given in the question.
3. Determine if the time period given is in the past or future.
4. Adjust the current date based on the time period and direction (past or future).
5. Convert the adjusted date to the required format (MM/DD/YYYY) as specified in the question.
6. Provide the final date as the answer to the question.

**Executing the strategy on a task example**:
**Question:** The concert was scheduled to be on 06/01/1943, but was delayed by one day to today. What is the date 10 days ago in MM/DD/YYYY?
**Solution:**
1. Determine the current date based on the given information.
- The concert was delayed by one day, so the current date is 06/02/1943.
2. Calculate the number of days given in the question.
- The question asks for the date 10 days ago.
3. Determine if the time period given is in the past or future.
- The time period is in the past.
4. Adjust the current date based on the time period and direction (past or future).
- Subtract 10 days from the current date: 06/02/1943 - 10 days = 05/23/1943.
5. Convert the adjusted date to the required format (MM/DD/YYYY) as specified in the question.
- The adjusted date is already in the required format: 05/23/1943.
6. Provide the final date as the answer to the question.
**Answer:** The answer is 05/23/1943.

Figure 24: Strategy generated by GPT-3.5 for Date Understanding and its execution on a task example.

**Word Sorting**

**Strategy**:
1. Identify the list of words that needs to be sorted.
2. Split the list of words into individual words.
3. Sort the individual words alphabetically.
4. Combine the sorted words back into a single list.
5. Present the sorted list of words as the answer.

**Executing the strategy on a task example**:
**Question:** Sort the following words alphabetically: List: oven costume counterpart
**Solution:**
1. Identify the list of words that needs to be sorted.
- The list of words is "oven costume counterpart".
2. Split the list of words into individual words.
- The individual words are "oven", "costume", and "counterpart".
3. Sort the individual words alphabetically.
- The sorted words are "costume", "counterpart", and "oven".
4. Combine the sorted words back into a single list.
- The combined sorted list is "costume counterpart oven".
5. Present the sorted list of words as the answer.
**Answer:** The answer is costume counterpart oven.

Figure 25: Strategy generated by GPT-3.5 for Word Sorting and its execution on a task example.

## Multi-step Arithmetic

**Strategy**:
1. Identify the innermost operations enclosed in parentheses.
2. Evaluate the innermost operations first.
3. Replace the evaluated expression with the result.
4. Repeat steps 1-3 until all parentheses are removed.
5. Evaluate the remaining arithmetic operations according to the order of operations (multiplication and division before addition and subtraction).
6. Evaluate all multiplication and division operations from left to right.
7. Replace the evaluated expression with the result.
8. Repeat steps 5-7 until there are no more multiplication and division operations to be performed.
9. Evaluate all addition and subtraction operations from left to right.
10. Replace the evaluated expression with the result.
11. Repeat steps 9-10 until there are no more addition and subtraction operations to be performed.
12. The final result is the solution to the multi-step arithmetic problem.

**Executing the strategy on a task example**:
**Question:** ((-9 * 7 * 7 * -9) + (4 * -9 - 8 - -4)) =
**Solution:**
Step 1: Identify the innermost operations enclosed in parentheses.
The innermost operation is (4 * -9 - 8 - -4).
Step 2: Evaluate the innermost operations first.
4 * -9 = -36
-36 - 8 = -44
-44 - -4 = -40
Step 3: Replace the evaluated expression with the result.
(((-9 * 7 * 7 * -9) + -40)
Step 4: Repeat steps 1-3 until all parentheses are removed.
The innermost operation is (-9 * 7 * 7 * -9).
Step 5: Evaluate the innermost operations first.
-9 * 7 = -63
-63 * 7 = -441
-441 * -9 = 3969
Step 6: Replace the evaluated expression with the result.
(3969 + -40)
Step 7: Repeat steps 5-6 until there are no more multiplication and division operations to be performed.
There are no more multiplication and division operations to be performed.
Step 8: Evaluate all addition and subtraction operations from left to right.
3969 + -40 = 3929
Step 9: Replace the evaluated expression with the result.
**Answer:** The answer is 3929.

Figure 26: Strategy generated by GPT-3.5 for Multi-step Arithmetic and its execution on a task example.

**Last Letter Concatenation**

**Strategy**:
1. Identify the given input, which will be a string containing multiple words separated by spaces.
2. Split the input string into individual words.
3. Initialize an empty string to store the last letters of the words.
4. Iterate through each word in the list of words.
5. Get the last letter of each word.
6. Concatenate the last letter to the string from step 3.
7. Continue steps 4-6 until all words have been processed.
8. Print or return the final concatenated string.

**Executing the strategy on a task example**:
**Question:** Take the last letters of the words in "Sergey Brin" and concatenate them.
**Solution:**
1. Identify the given input: "Sergey Brin"
2. Split the input string into individual words: ["Sergey", "Brin"]
3. Initialize an empty string: last_letters = ""
4. Iterate through each word in the list of words:
- For the word "Sergey":
- Get the last letter: last_letter = "y"
- Concatenate the last letter to the string: last_letters = "y"
- For the word "Brin":
- Get the last letter: last_letter = "n"
- Concatenate the last letter to the string: last_letters = "yn"
 5. Print or return the final concatenated string.
**Answer:** The answer is yn.

Figure 27: Strategy generated by GPT-3.5 for Last Letter Concatenation and its execution on a task example.

**MATH-Counting and Probability**

**Strategy**:
1. Identify the type of the counting or probability problem presented.
2. If the problem is regarding permutations or combinations, identify all possible outcomes and arrange or select the required number of objects accordingly.
3. If the problem is about probability, assess the sample space, the number of possible outcomes and the number of favorable outcomes.
4. For probability questions, use the formula of probability, which is the number of favorable outcomes divided by the number of total outcomes.
5. If the problem involves conditions or restrictions, ensure to factor in these when calculating probability or counting the possible arrangements or selections.
6. If the problem involves complex calculations, break them down into smaller manageable calculations.
7. Apply the suitable rule or formula according to the type of counting or probability problem, such as multiplication rule, addition rule, arrangement rules, rules of permutations and combinations, or probability rules.
8. Compute the result using arithmetic calculation or mathematical formulas, and write down the answer.
9. Review the process and the answer to ensure accuracy.

**Executing the strategy on a task example**:
**Question:** Given that $\binom{17}{7} = 19448$, $\binom{17}{8} = 24310$ and $\binom{17}{9} = 24310$, calculate $\binom{19}{9}$.
**Solution:**
1. The problem is about combinations, which is a type of counting problem.
2. The problem is asking for the number of ways to choose 9 items from a set of 19, which is represented by $\binom{19}{9}$.
3. We are given the values of $\binom{17}{7}$, $\binom{17}{8}$, and $\binom{17}{9}$, which can be used to calculate $\binom{19}{9}$.
4. The formula for combinations is $\binom{n}{r} = \frac{n!}{r!(n-r)!}$, where n is the total number of items, r is the number of items to choose, and '!' denotes factorial.
5. However, we can use the property of combinations that $\binom{n}{r} = \binom{n}{n-r}$, and the relationship between $\binom{n+1}{r}$ and $\binom{n}{r-1}$ and $\binom{n}{r}$, which is $\binom{n+1}{r} = \binom{n}{r-1} + \binom{n}{r}$.
6. Using these properties, we can express $\binom{19}{9}$ as $\binom{18}{8} + \binom{18}{9}$.
7. Similarly, $\binom{18}{8}$ can be expressed as $\binom{17}{7} + \binom{17}{8}$, and $\binom{18}{9}$ can be expressed as $\binom{17}{8} + \binom{17}{9}$.
8. Substituting the given values, we get $\binom{19}{9} = (19448 + 24310) + (24310 + 24310)$.
9. Performing the arithmetic calculation, we get $\binom{19}{9} = 43758 + 48620 = 92378$.
**Answer:** The answer is 92378.

Figure 28: Strategy generated by GPT-4 for MATH-Counting and Probability and its execution on a task example.

## StrategyQA

**Strategy**:
1. Identify the main subject(s) or concept(s) in the question.
2. Identify the relationship(s) or action(s) associated with the main subject(s) or concept(s).
3. Break down the question into sub-questions based on the identified subjects or concepts and their associated relationships or actions, ensuring each sub-question leads to additional information required to answer the main question.
4. For each sub-question, infer or retrieve relevant information. This can come from provided information, general knowledge, or external resources.
5. Compare and synthesize the information gathered from each sub-question.
6. Based on the synthesis, determine if the information satisfies or contradicts the main question.
7. Formulate a 'yes' or 'no' answer to the main question based on the evaluation.
8. Confirm that the answer accurately addresses the main question. If not, revisit the sub-questions and retrieval process.
9. Record the answer.

**Executing the strategy on a task example**:
**Question:** Could Brooke Shields succeed at University of Pennsylvania?
**Solution:**
1. The main subject in the question is Brooke Shields and the main concept is University of Pennsylvania.
2. The relationship associated with the main subject and concept is Brooke Shields' potential success at the University of Pennsylvania.
3. Sub-questions based on the identified subjects or concepts and their associated relationships or actions could be:
- Who is Brooke Shields?
- What is the University of Pennsylvania?
- What does it mean to succeed at the University of Pennsylvania?
4. For each sub-Question:
- Brooke Shields is a well-known American actress and model.
- The University of Pennsylvania is an Ivy League research university in Philadelphia. It is one of the most prestigious universities in the world.
- To succeed at the University of Pennsylvania would likely mean to graduate with a degree, possibly with honors or high academic achievement.
5. Comparing and synthesizing the information gathered from each sub-question, we know that Brooke Shields is a successful actress and model, and the University of Pennsylvania is a prestigious university. Success at the university would likely require dedication, intelligence, and hard work.
6. Based on this synthesis, it is possible that Brooke Shields could succeed at the University of Pennsylvania, given her demonstrated success in her career. However, without specific information about her academic abilities or aspirations, we cannot definitively say.
7. Given the information available, we can infer a 'yes' answer to the main question.
8. The answer 'yes' accurately addresses the main question, given the information available.
9. The answer is recorded.
**Answer:** The answer is yes.

Figure 29: Strategy generated by GPT-4 for StrategyQA and its execution on a task example.

---

**Multi-step Arithmetic**

**Strategy**:
1. Begin by identifying all the numbers and operations in the given problem.
2. Apply the mathematical rules of order operations. The correct order of operations is Parentheses, Exponents, Multiplication and Division (from left to right), Addition and Subtraction (from left to right). This rule is often abbreviated as PEMDAS.
3. Start solving the problem from the inside of the parentheses first, keeping PEMDAS rule in mind.
4. While doing multiplication, division, addition, and subtraction (in the order), make sure to change the sign of the number if there is a negative sign before it.
5. After solving the operations inside the parentheses, proceed solving the operations outside the parentheses. Remember to still follow the rules and order of operations.
6. Continue to calculate step by step until you reach the final answer.

**Executing the strategy on a task example**:
**Question:** ((-9 * 7 * 7 * -9) + (4 * -9 - 8 - -4)) =
**Solution:**
1. Identify all the numbers and operations in the given problem. The numbers are -9, 7, -9, 4, -9, 8, -4. The operations are multiplication (*), addition (+), and subtraction (-).
2. Apply the mathematical rules of order operations (PEMDAS).
3. Start solving the problem from the inside of the parentheses first.
- The first parentheses is (-9 * 7 * 7 * -9). According to the order of operations, we should do multiplication first. So, -9 * 7 = -63, -63 * 7 = -441, -441 * -9 = 3969.
- The second parentheses is (4 * -9 - 8 - -4). According to the order of operations, we should do multiplication first and then subtraction. So, 4 * -9 = -36, -36 - 8 = -44, -44 - -4 = -40.
4. After solving the operations inside the parentheses, proceed solving the operations outside the parentheses. So, 3969 + -40 = 3929.
5. Continue to calculate step by step until you reach the final answer.
**Answer:** The answer is 3929.

---

Figure 30: Strategy generated by GPT-4 for Multi-step Arithmetic and its execution on a task example.

## CoT Prompt for MATH-Algebra

**Question:** Calculate $\sqrt{75x} \cdot \sqrt{2x} \cdot \sqrt{14x}$ . Express your answer in simplest radical form in terms of $x$.
**Solution:** Writing everything in terms of prime factorizations, the given expression is $\sqrt{3 \cdot 5^2 \cdot 2 \cdot 2 \cdot 7 \cdot x^3} = \sqrt{(2^2 \cdot 5^2 \cdot x^2) \cdot (3 \cdot 7 \cdot x)} = 10x\sqrt{21x}$.
**Answer:** The answer is $10x\sqrt{21x}$

**Question:** Rationalize the denominator of $\frac{\sqrt{32}}{\sqrt{16}-\sqrt{2}}$. The answer can be written as $\frac{A\sqrt{B}+C}{D}$, where $A$, $B$, $C$, and $D$ are integers, $D$ is positive, and $B$ is not divisible by the square of any prime. Find the minimum possible value of $A + B + C + D$.
**Solution:** We can write the numerator as $4\sqrt{2}$, the denominator as $4 - \sqrt{2}$. Then, we multiply the numerator and denominator by the conjugate of the denominator.

$$\frac{4\sqrt{2}}{4-\sqrt{2}} \cdot \frac{4+\sqrt{2}}{4+\sqrt{2}} =$$

$$\frac{16\sqrt{2}+8}{14} =$$

$$\frac{8\sqrt{2}+4}{7}$$

Therefore, $A + B + C + D = 8 + 2 + 4 + 7 = 21$.
**Answer:** The answer is 21

**Question:** The quadratic $8x^2 + 12x - 14$ has two real roots. What is the sum of the squares of these roots? Express your answer as a common fraction in lowest terms.
**Solution:** Let $x_1$ and $x_2$ be the roots of the equation $8x^2 + 12x - 14$. We want to find $x_1^2 + x_2^2$. Note that $x_1^2 + x_2^2 = (x_1 + x_2)^2 - 2x_1x_2$. We know that $x_1 + x_2$, the sum of the roots, is equal to $\frac{-b}{a}$, which for this equation is $\frac{-12}{8} = \frac{-3}{2}$. Likewise, we know that $x_1x_2$, the product of the roots, is equal to $\frac{c}{a}$, which for this equation is $\frac{-14}{8} = \frac{-7}{4}$. Thus, $x_1^2 + x_2^2 = \left(\frac{-3}{2}\right)^2 - 2\left(\frac{-7}{4}\right) = \frac{9}{4} + \frac{14}{4} = \frac{23}{4}$.
**Answer:** The answer is $\frac{23}{4}$

**Question:** What is the product of all the coordinates of all the points of intersection of the two circles defined by $x^2 - 2x + y^2 - 10y + 25 = 0$ and $x^2 - 8x + y^2 - 10y + 37 = 0$?
**Solution:** Add $(-2/2)^2$ and $(-10/2)^2$ to the first equation and $(-8/2)^2$ and $(-10/2)^2$ to the second equation to find that the given equations are equivalent to

$$(x^2 - 2x + 1) + (y^2 - 10y + 25) = 1, \text{ and}$$
$$(x^2 - 8x + 16) + (y^2 - 10y + 25) = 4$$

which are equivalent to

$$(x - 1)^2 + (y - 5)^2 = 1^2,$$
$$(x - 4)^2 + (y - 5)^2 = 2^2,$$

respectively. Hence, the two circles have centers $(1, 5)$ and $(4, 5)$ respectively and radii 1 and 2 respectively. Since the centers of the circles are 3 units apart and the sum of their radii is 3, the two circles intersect at only one point. We can see that $(2, 5)$ is the desired intersection point, so our product is $2 \cdot 5 = 10$.
**Answer:** The answer is 10

Figure 31: Chain-of-thought (CoT) prompt for MATH-Algebra.

**Question:** Pete thinks of a number. He doubles it, adds 10, multiplies by 4, and ends up with 120. What was his original number?
**Solution:** Let Pete's original number be $x$. If he double it and adds 10, he will have $2x + 10$. After multiplying by 4, Pete ended up with 120. From this information, we have the equation: $4(2x + 10) = 120$. Expanding the left side and solving, we find:

$$8x + 40 = 120$$
$$\Rightarrow \quad 8x = 80$$
$$\Rightarrow \quad x = 10.$$

**Answer:** The answer is 10

**Question:** A standard deck of 52 cards has 13 ranks (Ace, 2, 3, 4, 5, 6, 7, 8, 9, 10, Jack, Queen, King) and 4 suits ($\spadesuit$, $\heartsuit$, $\diamondsuit$, and $\clubsuit$), such that there is exactly one card for any given rank and suit. Two of the suits ($\spadesuit$ and $\clubsuit$) are black and the other two suits ($\heartsuit$ and $\diamondsuit$) are red. The deck is randomly arranged. What is the probability that the top card is a $\heartsuit$?
**Solution:** There are 13 $\heartsuit$'s and 52 cards total, so the probability that the top card is a $\heartsuit$ is $\dfrac{13}{52} = \dfrac{1}{4}$.
**Answer:** The answer is $\frac{1}{4}$

**Question:** A number in the set $\{50, 51, 52, 53, ..., 999\}$ is randomly selected. What is the probability that it is a two-digit number? Express your answer as a common fraction.
**Solution:** To count the number of numbers in this set, we subtract 49 from all of the numbers, giving the set $\{1, 2, 3, \ldots, 950\}$, making it obvious that there are 950 numbers total. Furthermore, the set $\{50, 51, 52, \ldots, 98, 99\}$ corresponds to the more easily counted $\{1, 2, 3, \ldots, 49, 50\}$ by subtracting 49. So, the probability of selecting a two-digit number is $\frac{50}{950} = \frac{1}{19}$.
**Answer:** The answer is $\frac{1}{19}$

**Question:** A pentagon is drawn by placing an isosceles right triangle on top of a square as pictured. What percent of the area of the pentagon is the area of the right triangle?
[asy] size(50); draw((0,0)–(0,-1)–(1,-1)–(1,0)–(0,0)–(.5,.5)–(1,0)); [/asy]
**Solution:** Let the leg length of the isosceles right triangle be $x$, so the hypotenuse of the triangle has length $x\sqrt{2}$. The hypotenuse of the triangle is a side of the square, so the area of the square is $(x\sqrt{2})^2 = 2x^2$. The area of the triangle is $(x)(x)/2 = x^2/2$. So, the area of the pentagon is

$$\frac{x^2}{2} + 2x^2 = \frac{5x^2}{2}.$$

Therefore, the fraction of the pentagon's area that is inside the triangle is

$$\frac{x^2/2}{5x^2/2} = \frac{x^2}{2} \cdot \frac{2}{5x^2} = \frac{1}{5} = 20\%.$$

**Answer:** The answer is 20

Figure 32: Chain-of-thought (CoT) prompt for MATH-Prealgebra.

**Question:** Find the quadratic polynomial $p(x)$ such that $p(-3) = 10$, $p(0) = 1$, and $p(2) = 5$.
**Solution:** Let $p(x) = ax^2 + bx + c$. Then from the given information,

$$9a - 3b + c = 10,$$
$$c = 1,$$
$$4a + 2b + c = 5.$$

Then $9a - 3b = 9$ and $4a + 2b = 4$, which reduce to $3a - b = 3$ and $2a + b = 2$. Adding, we get $5a = 5$, so $a = 1$. Then $4 + 2b = 4$, so $b = 0$. Therefore, $p(x) = x^2 + 1$.
**Answer:** The answer is $x^2 + 1$

**Question:** Calculate: $\frac{5}{3} \times \frac{6}{10} \times \frac{15}{9} \times \frac{12}{20} \times \frac{25}{15} \times \frac{18}{30} \times \frac{35}{21} \times \frac{24}{40}$
**Solution:** Each of the fractions $\frac{5}{3}, \frac{15}{9}, \frac{25}{15}, \frac{35}{21}$ reduce to $\frac{5}{3}$, and each of the fractions $\frac{6}{10}, \frac{12}{20}, \frac{18}{30}, \frac{24}{40}$ reduce to $\frac{3}{5}$. Therefore, the product of all eight fractions is 1.
**Answer:** The answer is 1

**Question:** For $-25 \le x \le 25$, find the maximum value of $\sqrt{25 + x} + \sqrt{25 - x}$.
**Solution:** By QM-AM,

$$\frac{\sqrt{25 + x} + \sqrt{25 - x}}{2} \le \sqrt{\frac{25 + x + 25 - x}{2}} = 5,$$

so $\sqrt{25 + x} + \sqrt{25 - x} \le 10$. Equality occurs at $x = 0$, so the maximum value is 10.
**Answer:** The answer is 10

**Question:** The hyperbolas

$$\frac{x^2}{4} - \frac{y^2}{9} = 1$$

and

$$\frac{y^2}{18} - \frac{x^2}{N} = 1$$

have the same asymptotes. Find $N$.
**Solution:** In general, for the hyperbola $\frac{x^2}{a^2} - \frac{y^2}{b^2} = 1$, the asymptotes are $\frac{x}{a} = \pm\frac{y}{b}$, or $y = \pm\frac{b}{a}x$. Therefore, the asymptotes of the first hyperbola are $y = \pm\frac{3}{2}x$.

For the hyperbola $\frac{y^2}{a^2} - \frac{x^2}{b^2} = 1$, the asymptotes are $\frac{y}{a} = \pm\frac{x}{b}$, or $y = \pm\frac{a}{b}x$. Therefore, the asymptotes of the second hyperbola are $y = \pm\frac{3\sqrt{2}}{\sqrt{N}}x$.

For the two hyperbolas to have the same asymptotes, we must have $\frac{3}{2} = \frac{3\sqrt{2}}{\sqrt{N}}$. Solving for $N$ gives $N = 8$.
[asy] void axes(real x0, real x1, real y0, real y1) { draw((x0,0)–(x1,0),EndArrow); draw((0,y0)–(0,y1),EndArrow); label("$x$",(x1,0),E); label("$y$",(0,y1),N); for (int i=floor(x0)+1; i<x1; ++i) draw((i,.1)–(i,-.1)); for (int i=floor(y0)+1; i<y1; ++i) draw((.1,i)–(-.1,i)); } path[] yh(real a, real b, real h, real k, real x0, real x1, bool upper=true, bool lower=true, pen color=black) { real f(real x) return k + a/b * sqrt(b^2 + (x - h)^2); real g(real x) return k - a/b * sqrt(b^2 + (x - h)^2); if (upper) draw(graph(f, x0, x1),color, Arrows); if (lower) draw(graph(g, x0, x1),color, Arrows); path [] arr = graph(f, x0, x1), graph(g, x0, x1); return arr; } void xh(real a, real b, real h, real k, real y0, real y1, bool right=true, bool left=true, pen color=black) { path [] arr = yh(a, b, k, h, y0, y1, false, false); if (right) draw(reflect((0,0),(1,1))*arr[0],color, Arrows); if (left) draw(reflect((0,0),(1,1))*arr[1],color, Arrows); } void e(real a, real b, real h, real k) { draw(shift((h,k))*scale(a,b)*unitcircle); } size(8cm); axes(-8,8, -10, 10); xh(2, 3, 0, 0, -8, 8); yh(3*sqrt(2),sqrt(8),0,0,-5,5); draw((-6,9)–(6,-9)h6,9)–(-6,-9),dotted); [/asy]
**Answer:** The answer is 8

Figure 33: Chain-of-thought (CoT) prompt for MATH-Intermediate Algebra.

## CoT Prompt for MATH-Counting and Probability

**Question:** The digits 2, 3, 5 and 7 are arranged randomly to form a four-digit number. What is the probability that the number is odd? Express your answer as a common fraction.

**Solution:** The number formed is odd if and only if its units digit is not 2. Since the digits 2, 3, 5, and 7 are arranged randomly, the probability that 2 is the units digit is 1/4. Therefore, the probability that the number is odd is $1 - 1/4 = \frac{3}{4}$.

**Answer:** The answer is $\frac{3}{4}$

**Question:** Eight congruent equilateral triangles, each of a different color, are used to construct a regular octahedron. How many distinguishable ways are there to construct the octahedron? (Two colored octahedrons are distinguishable if neither can be rotated to look just like the other.)

[asy] import three; import math; unitsize(1.5cm); currentprojection=orthographic(2,0.2,1); triple A=(0,0,1); triple B=(sqrt(2)/2,sqrt(2)/2,0); triple C=(sqrt(2)/2,-sqrt(2)/2,0); triple D=(-sqrt(2)/2,-sqrt(2)/2,0); triple E=(-sqrt(2)/2,sqrt(2)/2,0); triple F=(0,0,-1); draw(A–B–E–cycle); draw(A–C–D–cycle); draw(F–C–B–cycle); draw(F–D–E–cycle,dotted+linewidth(0.7)); [/asy]

**(A)** 210     **(B)** 560     **(C)** 840     **(D)** 1260     **(E)** 1680

**Solution:** Since the octahedron is indistinguishable by rotations, without loss of generality fix a face to be red.

[asy] size(8cm); defaultpen(0.5); import three; import math; currentprojection=orthographic(2,0.2,1); triple A=(0,0,1); triple B=(sqrt(2)/2,sqrt(2)/2,0); triple C=(sqrt(2)/2,-sqrt(2)/2,0); triple D=(-sqrt(2)/2,-sqrt(2)/2,0); triple E=(-sqrt(2)/2,sqrt(2)/2,0); triple F=(0,0,-1); draw(A–B–E–cycle); draw(A–C–D–cycle); draw(F–C–B–cycle); draw(F–D–E–cycle,dotted+linewidth(0.7)); draw(surface(A–B–C–cycle),rgb(1,.6,.6),nolight);[/asy] There are 7! ways to arrange the remaining seven colors, but there still are three possible rotations about the fixed face, so the answer is $7!/3 = 1680$.

[asy] size(8cm); defaultpen(0.5); import three; import math; currentprojection=orthographic(2,0,1); triple A=(0,0,1); triple B=(sqrt(2)/2,sqrt(2)/2,0); triple C=(sqrt(2)/2,-sqrt(2)/2,0); triple D=(-sqrt(2)/2,-sqrt(2)/2,0); triple E=(-sqrt(2)/2,sqrt(2)/2,0); triple F=(0,0,-1); triple right=(0,1,0); picture p = new picture, r = new picture, s = new picture; draw(p,A–B–E–cycle); draw(p,A–C–D–cycle); draw(p,F–C–B–cycle); draw(p,F–D–E–cycle,dotted+linewidth(0.7)); draw(p,surface(A–B–C–cycle),rgb(1,.6,.6),nolight); draw(p,surface(A–B–E–cycle),rgb(1,1,.6),nolight); add(scale3(2.2)*p); draw(r,A–B–E–cycle); draw(r,A–C–D–cycle); draw(r,F–C–B–cycle); draw(r,F–D–E–cycle,dotted+linewidth(0.7)); draw(r,surface(A–B–C–cycle),rgb(1,.6,.6),nolight); draw(r,surface(A–C–D–cycle),rgb(1,1,.6),nolight); add(scale3(2.2)*shift(2*right)*r); draw(s,A–B–E–cycle); draw(s,A–C–D–cycle); draw(s,F–C–B–cycle); draw(s,F–D–E–cycle,dotted+linewidth(0.7)); draw(s,surface(A–B–C–cycle),rgb(1,.6,.6),nolight); draw(s,surface(B–C–F–cycle),rgb(1,1,.6),nolight); add(scale3(2.2)*shift(4*right)*s); [/asy]

**Answer:** The answer is 1680

**Question:** The Gnollish language consists of 3 words, "splargh," "glumph," and "amr." In a sentence, "splargh" cannot come directly before "glumph"; all other sentences are grammatically correct (including sentences with repeated words). How many valid 3-word sentences are there in Gnollish?

**Solution:** We proceed by counting the complement, or the number of invalid 3-word sentences. A sentence is invalid precisely when it is of the form "(word) splargh glumph" or "splargh glumph (word)." There are 3 choices for the missing word in each sentence, and since each case is exclusive, we have a total of 6 invalid sentences. Since there are $3 \cdot 3 \cdot 3 = 27$ possible 3-word sentences with no restrictions, there are $27 - 6 = 21$ that satisfy the restrictions of the problem.

**Answer:** The answer is 21

**Question:** Given that $\binom{17}{7} = 19448$, $\binom{17}{8} = 24310$ and $\binom{17}{9} = 24310$, calculate $\binom{19}{9}$.

**Solution:** We can apply Pascal's identity to get that $\binom{19}{9} = \binom{18}{8} + \binom{18}{9}$. From here, we can apply it twice more to get that $\binom{19}{9} = \binom{18}{8} + \binom{18}{9} = \left(\binom{17}{7} + \binom{17}{8}\right) + \left(\binom{17}{8} + \binom{17}{9}\right)$. Substituting the provided values of $\binom{17}{7}$, $\binom{17}{8}$, and $\binom{17}{9}$ gives us $\binom{19}{9} = 19448 + 2(24310) + 24310 = 92378$.

**Answer:** The answer is 92378

Figure 34: Chain-of-thought (CoT) prompt for MATH-Counting and Probability.

**Question:** Let $x$ be a positive integer such that $9x \equiv 1 \pmod{25}$. What is the remainder when $11 + x$ is divided by 25?

**Solution:** The given information can be expressed by writing $x \equiv 9^{-1} \pmod{25}$. Thus we wish to compute $11 + 9^{-1} \pmod{25}$.

Modulo 25, we can write 11 as $11 \cdot (9 \cdot 9^{-1}) \equiv (11 \cdot 9) \cdot 9^{-1} \equiv 99 \cdot 9^{-1}$. Thus

$$11 + 9^{-1} \equiv 99 \cdot 9^{-1} + 1 \cdot 9^{-1}$$
$$\equiv 100 \cdot 9^{-1}$$
$$\equiv 0 \cdot 9^{-1}$$
$$\equiv 0 \pmod{25},$$

so the remainder when $11 + x$ is divided by 25 is 0.

Notice that the trick we used here is analogous to using a common denominator to add fractions.

**Answer:** The answer is 0

**Question:** Let $m$ be the product of all positive integers less than $4!$ which are invertible modulo $4!$. Find the remainder when $m$ is divided by $4!$.(Here $n!$ denotes $1 \times \cdots \times n$ for each positive integer $n$.)

**Solution:** We compute that $4! = 1 \times 2 \times 3 \times 4 = 2^3 \times 3 = 24$. So we want exactly the numbers in the set $\{1, \ldots, 24\}$ which are divisible by neither 2 nor 3, since an integer $a$ is invertible modulo $n$ for some positive integer $n$ if and only if $\gcd(a, n) = 1$. These turn out to be $\{1, 5, 7, 11, 13, 17, 19, 23\}$. Then

$$m \equiv 1 \cdot 5 \cdot 7 \cdot 11 \cdot 13 \cdot 17 \cdot 19 \cdot 23$$
$$\equiv 1 \cdot 5 \cdot 7 \cdot 11 \cdot (-11) \cdot (-7) \cdot (-5) \cdot (-1)$$
$$\equiv (5 \cdot 7 \cdot 11)^2$$
$$\equiv (35 \cdot 11)^2$$
$$\equiv (11 \cdot 11)^2$$
$$\equiv (121)^2$$
$$\equiv 1^2$$
$$\equiv 1 \pmod{24}$$

**Answer:** The answer is 1

**Question:** Find $2^{-1} \pmod{185}$, as a residue modulo 185. (Give an answer between 0 and 184, inclusive.)

**Solution:** Since $2 \cdot 93 \equiv 186 \equiv 1 \pmod{185}$, $2^{-1} \equiv 93 \pmod{185}$.

**Answer:** The answer is 93

**Question:** What is the remainder when 2007 is divided by 81?

**Solution:** Dividing using long division, we find that $2007 = 81 \cdot 24 + 63$, so the remainder is 63.

**Answer:** The answer is 63

Figure 35: Chain-of-thought (CoT) prompt for MATH-Number Theory.

**Question:** Two sectors of a circle of radius 12 overlap as shown, with $P$ and $R$ as the centers of the respective circles. Determine the area of the shaded region.

[asy] draw((0,0)–(10.3923,-6)–(20.7846,0)–(10.3923,6)–cycle,black+linewidth(1)); filldraw((10.3923,6)..(12,0)..(10.3923,-6)–cycle,gray,black+linewidth(1)); filldraw((10.3923,6)..(8.7846,0)..(10.3923,-6)–cycle,gray,black+linewidth(1)); label("$P$",(0,0),W); label("$Q$",(10.3923,6),N); label("$R$",(20.7846,0),E); label("$S$",(10.3923,-6),S); label("$60°$",(0,0),2E); label("$60°$",(20.7846,0),2W); [/asy]

**Solution:** By symmetry, the areas of the two parts of the shaded region are equal. Consider the right part of the shaded region and the left triangle.

[asy] draw((0,0)–(10.3923,-6)–(10.3923,6)–cycle,black+linewidth(1)); filldraw((10.3923,6)..(12,0)..(10.3923,-6)–cycle,gray,black+linewidth(1)); draw((0,0)–(10.3923,0),black+linewidth(1)); draw((10.3923,0)–(9.3923,0)–(9.3923,1)–(10.3923,1),black+linewidth(1)); label("$P$",(0,0),W); label("$Q$",(10.3923,6),N); label("$S$",(10.3923,-6),S); label("$Z$",(10.3923,0),SW); [/asy]

The shaded area is equal to the area of sector $PQS$ minus the area of triangle $PQS$.

Since $\angle PQS = 60°$ and $PQ = 12$, the area of sector $PQS$ is

$$\frac{1}{6} \cdot 12^2 \cdot \pi = 24\pi.$$

Also, triangle $PQS$ is equilateral with side length 12, so its area is

$$\frac{\sqrt{3}}{4} \cdot 12^2 = 36\sqrt{3}.$$

Thus, the area of the right part of the shaded region is $24\pi - 36\sqrt{3}$, so the area of the entire shaded region is

$$2(24\pi - 36\sqrt{3}) = 48\pi - 72\sqrt{3}.$$

**Answer:** The answer is $48\pi - 72\sqrt{3}$

**Question:** The square with vertices $(-a, -a), (a, -a), (-a, a), (a, a)$ is cut by the line $y = x/2$ into congruent quadrilaterals. The perimeter of one of these congruent quadrilaterals divided by $a$ equals what? Express your answer in simplified radical form.

**Solution:** The line $y = \frac{x}{2}$ will intersect the two vertical sides of the square, as shown below:

[asy] real f(real x) { return x/2; } import graph; size(6cm); real a = 8; pair A=(-a,a), B=(a,a), C=(a,-a), D=(-a,-a); draw(A–B–C–D–cycle); draw(graph(f,-11,11),Arrows); axes(Arrows(4)); dot("$(-a, a)$",A,N); dot("$(a, a)$",B,N); dot("$(a, -a)$",C,S); dot("$(-a, -a)$",D,S); real eps=0.2; dot((8,4)h-8,-4)); draw(shift((10,0))*"$2a$",(-a+eps,-a/2-.5)–(a-eps,-a/2-.5),Arrows); draw(shift((0,10))*"$a$",(a+2*eps,-a/2)–(a+2*eps,a/2),Arrows);[/asy] The equation of the right side of the square is $x = a$, so we have $y = \frac{x}{2} = \frac{a}{2}$, which means that the intersection point with the right side of the square is $\left(a, \frac{a}{2}\right)$. Similarly, the equation of the left side of the square is $x = -a$, so we have $y = \frac{x}{2} = -\frac{a}{2}$, which means that the intersection point with the left side of the square is $\left(-a, -\frac{a}{2}\right)$. It follows that the sides of each quadrilateral have lengths $\frac{a}{2}$, $2a$, $\frac{3a}{2}$, and $\sqrt{a^2 + (2a)^2} = a\sqrt{5}$, by the Pythagorean theorem. Hence, the perimeter of the quadrilateral is

$$\frac{a}{2} + 2a + \frac{3a}{2} + a\sqrt{5} = \left(4 + \sqrt{5}\right)a,$$

and when this is divided by $a$, we get $4 + \sqrt{5}$.

**Answer:** The answer is $4 + \sqrt{5}$

Figure 36: Chain-of-thought (CoT) prompt for MATH-Geometry (Part 1).

**Question:** A right circular cylinder with radius 2 is inscribed in a hemisphere with radius 5 so that its bases are parallel to the base of the hemisphere. What is the height of this cylinder?

**Solution:** We draw and label a diagram as follows:

[asy] size(110); pair O = (0,0); pair A = (.3,.94); pair B = (.3,.075); draw(O–A–B–cycle,heavycyan); label("$O$",O,W); label("$A$",A,N); label("$B$",B,S); import solids; import three; defaultpen(linewidth(0.8)); currentprojection = orthographic(5,0,1.3); revolution c = cylinder((0,0,0), .4, .91); draw(c,black); draw(scale(1,.25)*arc((0,0),1,0,180),dashed); draw(scale(1,.25)*arc((0,0),1,180,360)); draw(Arc((0,0),1,0,180)); [/asy]

Let the center of the hemisphere be $O$, and let $A$ be a point on the circumference of the top circle of the cylinder. Since the cylinder is inscribed in the hemisphere, $A$ lies on the hemisphere as well, so $OA = 5$. We drop a perpendicular from $A$ to the base of the hemisphere and let it intersect the base of the hemisphere at $B$. Since the cylinder is right and $AB$ is a height of the cylinder, $\angle OBA$ is a right angle, and $B$ lies on the circumference of the bottom circle of the cylinder. Thus, $OB$ is a radius of the cylinder, so $OB = 2$. We have that $\triangle OBA$ is right, so by the Pythagorean theorem, we have

$$AB = \sqrt{OA^2 - OB^2} = \sqrt{5^2 - 2^2} = \sqrt{21}.$$

Thus, the height of the cylinder is $\sqrt{21}$.

**Answer:** The answer is $\sqrt{21}$

**Question:** Parallelogram $ABCD$ with $A(2,5)$, $B(4,9)$, $C(6,5)$, and $D(4,1)$ is reflected across the $x$-axis to $A'B'C'D'$ and then $A'B'C'D'$ is reflected across the line $y = x + 1$ to $A''B''C''D''$. This is done such that $D'$ is the image of $D$, and $D''$ is the image of $D'$. What is the ordered pair of $D''$ in the coordinate plane?

**Solution:** Reflecting a point across the $x$-axis multiplies its $y$-coordinate by $-1$. Therefore, $D' = (4,-1)$. To reflect $D'$ across the line $y = x + 1$, we first translate both the line and the point down one unit so that the equation of the translated line is $y = x$ and the coordinates of the translated point are $(4,-2)$. To reflect across $y = x$, we switch the $x$-coordinate and $y$-coordinate to obtain $(-2,4)$. Translating this point one unit up, we find that $D'' = (-2,5)$.

**Answer:** The answer is (-2,5)

Figure 37: Chain-of-thought (CoT) prompt for MATH-Geometry (Part 2).

**Question:** The function

$$f(z) = \frac{(-1 + i\sqrt{3})z + (-2\sqrt{3} - 18i)}{2}$$

represents a rotation around some complex number $c$. Find $c$.

**Solution:** Since a rotation around $c$ fixes $c$, the complex number $c$ must satisfy $f(c) = c$. In other words,

$$c = \frac{(-1 + i\sqrt{3})c + (-2\sqrt{3} - 18i)}{2}$$

Then $2c = (-1 + i\sqrt{3})c + (-2\sqrt{3} - 18i)$, so

$$(3 - i\sqrt{3})c = -2\sqrt{3} - 18i.$$

Then

$$\begin{aligned}
c &= \frac{-2\sqrt{3} - 18i}{3 - i\sqrt{3}} \\
&= \frac{(-2\sqrt{3} - 18i)(3 + i\sqrt{3})}{(3 - i\sqrt{3})(3 + i\sqrt{3})} \\
&= \frac{-6\sqrt{3} - 6i - 54i + 18\sqrt{3}}{12} \\
&= \frac{12\sqrt{3} - 60i}{12} \\
&= \sqrt{3} - 5i.
\end{aligned}$$

**Answer:** The answer is $\sqrt{3} - 5i$

**Question:** Convert the point $\left(8, \frac{7\pi}{6}\right)$ in polar coordinates to rectangular coordinates.

**Solution:** In rectangular coordinates, $\left(8, \frac{7\pi}{6}\right)$ becomes

$$\left(8\cos\frac{7\pi}{6}, 8\sin\frac{7\pi}{6}\right) = (-4\sqrt{3}, -4).$$

**Answer:** The answer is $(-4\sqrt{3}, -4)$

Figure 38: Chain-of-thought (CoT) prompt for MATH-Precalculus (Part 1).

## CoT Prompt for MATH-Precalculus (Part 2)

**Question:** Compute $\begin{pmatrix} 2 & 0 \\ 5 & -3 \end{pmatrix} \begin{pmatrix} 8 & -2 \\ 1 & 1 \end{pmatrix}$.

**Solution:** We have that

$$\begin{pmatrix} 2 & 0 \\ 5 & -3 \end{pmatrix} \begin{pmatrix} 8 & -2 \\ 1 & 1 \end{pmatrix} = \begin{pmatrix} (2)(8) + (0)(1) & (2)(-2) + (0)(1) \\ (5)(8) + (-3)(1) & (5)(-2) + (-3)(1) \end{pmatrix} = \begin{pmatrix} 16 & -4 \\ 37 & -13 \end{pmatrix}.$$

**Answer:** The answer is $\begin{pmatrix} 16 & -4 \\ 37 & -13 \end{pmatrix}$

**Question:** A line is parameterized by a parameter $t$, so that the vector on the line at $t = 2$ is $\begin{pmatrix} 1 \\ 4 \end{pmatrix}$, and the vector on the line at $t = 3$ is $\begin{pmatrix} 3 \\ -4 \end{pmatrix}$. Find the vector on the line at $t = -7$.

**Solution:** Let the line be

$$\begin{pmatrix} x \\ y \end{pmatrix} = \mathbf{a} + t\mathbf{d}.$$

Then from the given information,

$$\begin{pmatrix} 1 \\ 4 \end{pmatrix} = \mathbf{a} + 2\mathbf{d},$$

$$\begin{pmatrix} 3 \\ -4 \end{pmatrix} = \mathbf{a} + 3\mathbf{d}.$$

We can treat this system as a linear set of equations in $\mathbf{a}$ and $\mathbf{d}$. Accordingly, we can solve to get $\mathbf{a} = \begin{pmatrix} -3 \\ 20 \end{pmatrix}$ and $\mathbf{d} = \begin{pmatrix} 2 \\ -8 \end{pmatrix}$. Hence,

$$\begin{pmatrix} x \\ y \end{pmatrix} = \begin{pmatrix} -3 \\ 20 \end{pmatrix} + t \begin{pmatrix} 2 \\ -8 \end{pmatrix}.$$

Taking $t = -7$, we get

$$\begin{pmatrix} x \\ y \end{pmatrix} = \begin{pmatrix} -3 \\ 20 \end{pmatrix} - 7 \begin{pmatrix} 2 \\ -8 \end{pmatrix} = \begin{pmatrix} -17 \\ 76 \end{pmatrix}.$$

**Answer:** The answer is $\begin{pmatrix} -17 \\ 76 \end{pmatrix}$

Figure 39: Chain-of-thought (CoT) prompt for MATH-Precalculus (Part 2).

