# OpenReview forum: "StrategyLLM: Large Language Models as Strategy Generators, Executors, Optimizers, and Evaluators for Problem Solving"
_NeurIPS.cc/2024/Conference — NeurIPS 2024 poster_

### Official Review · Reviewer_D2vF · 2024-07-07

**Soundness:** 1
**Presentation:** 2
**Contribution:** 1
**Rating:** 5
**Confidence:** 3

**Summary:**

The paper proposes a prompt engineering method to address generalizability and consistency issues in existing prompting approaches for LLM-based problem solving. It utilizes four LLM-based agents including strategy generator, executor, optimizer, and evaluator. The proposed method works by generalizing knowledge from instances through induction and applying generalized knowledge to solve a given problem through deduction.
Authors evaluate the performance of their method empirically on multiple data sets and against a few alternative methods including self consistency with chain of thoughts. Results show superior performance for the proposed method compared with a few alternative approaches.

**Strengths:**

Authors have evaluated their method's performance across multiple data sets and against various alternative methods. In addition, they conduct experiments to assess the universality of their proposed method, the effectiveness of task level strategy for reasoning. They also analyze the cost of strategy-based prompt engineering.

**Weaknesses:**

The main gap in the paper is the lack of detailed explanation to bridge between the superior performance of the proposed method and other studied methods including CoT through concrete examples and clear intuitions. The intuition explained in the paper relies on generalization and consistency as the main traits of the proposed method. It would have been helpful to use a concrete example showing how alternative methods such as CoT fail to provide the right solution for a given problem while the  method succeeds at doing so. It could be argued that the raw information is already available in the Appendix section, however, the reviewer is asking for interpretation of the way CoT is solving a problem vs. the proposed method by highlighting the generalization and consistency gaps in CoT's approach.

**Questions:**

- Where does the feedback in strategy optimizer come from? Is it automatically generated by LLM? If yes, can you elaborate more on how it's generated? If not, can authors clarify where feedback is coming from?
- Authors report biggest gain from their method for symbolic reasoning tasks, but Addendum does not provide any examples of such tasks and the corresponding generated strategies.
- How does strategy LLM generate multiple sets of strategy? Is it by asking it to re-run the same prompt multiple times or does it involve changing the original prompt?

**Limitations:**

Authors have listed lack of domain specific knowledge in underlying LLMs due to lack of domain specific training data as a limitation of their work. Beyond this limitation, strategy LLM might be susceptible to bias by generating discriminating strategies that may perform well on the train set which is not addressed in the paper.

---

> ### Author Rebuttal · Authors · 2024-08-07
>
> Thank you for your insightful comments. We address each question as follows:
>
> > The intuition explained in the paper relies on generalization and consistency as the main traits of the proposed method. It would have been helpful to use a concrete example showing how alternative methods such as CoT fail to provide the right solution for a given problem while the method succeeds at doing so.
>
> We use an example from the test set of the CP dataset in the MATH benchmark for illustration.
>
> **Question**: Suppose that I have 6 different books, 2 of which are math books.  In how many ways can I stack my 6 books on a shelf if I want a math book on both ends of the stack?
>
> **Gold Answer**: 48
>
> The CoT prompt for the CP dataset is illustrated in Figure 33 on Page 36 of our paper, while the strategy generated by GPT-3.5 is detailed in Figure 18 on Page 22. The predictions of CoT and StrategyLLM are as follows:
>
> **CoT Prediction**:
> ```
> We can treat the two math books as a single entity. So we have 5 entities to arrange on the shelf, which can be done in $5!$ ways. However, within the math book entity, the two math books can be arranged in $2!$ ways. Therefore, the total number of ways to arrange the books is $5! \cdot 2! = \boxed{240}$.
> Answer: The answer is 240.
> ```
>
> **StrategyLLM Prediction**:
> ```
> 1. Understand the problem statement:
>   - We have 6 different books, 2 of which are math books.
>   - We need to stack the books on a shelf in a way that a math book is on both ends of the stack.
>
> 2. Identify the counting or probability concept:
>   - We need to use the concept of permutations to solve this problem.
>
> 3. Break down the problem:
>   - We have 6 books, 2 of which are math books.
>   - We need to arrange these books on a shelf in a way that a math book is on both ends of the stack.
>   - The math books can be arranged in 2 ways (one on the left end and one on the right end).
>   - The remaining 4 books can be arranged in 4! = 4 x 3 x 2 x 1 = 24 ways.
>
> 4. Calculate the required counting or probability:
>   - To calculate the total number of ways to stack the books, we need to multiply the number of ways to arrange the math books with the number of ways to arrange the remaining books.
>   - Total number of ways = 2 (ways to arrange math books) x 24 (ways to arrange remaining books) = 48.
>
> 5. Simplify or reduce the problem:
>   - There are no further simplifications or reductions possible for this problem.
>
> 6. Communicate the solution clearly:
>   - There are 48 ways to stack the 6 books on a shelf such that a math book is on both ends of the stack.
>
> Answer: The answer is 48.
> ```
>
> The CoT prediction fails to account for the requirement of placing a math book at both ends, resulting in an incorrect solution. In contrast, the generalizable strategy employed by StrategyLLM allows for a comprehensive understanding of the problem, effective utilization of relevant concepts, and a breakdown of the task. This approach ensures that critical details are not overlooked, leading to the correct answer. Additionally, consistently applying a strategy across few-shot examples illustrates how that strategy can be utilized in various contexts and ensures that the same strategy will be applied to the test instance.
>
> Beyond accuracy, the solutions derived from StrategyLLM are more detailed, interpretable, and accessible, making them particularly suitable for contexts that demand clarity and comprehensibility, such as educational settings.
>
> > Where does the feedback in strategy optimizer come from? Is it automatically generated by LLM? If yes, can you elaborate more on how it's generated? If not, can authors clarify where feedback is coming from?
>
> The feedback in the strategy optimizer is indeed automatically generated by LLMs. Following the execution of a strategy, we can identify which examples yield correct solutions and which do not. We then prompt the LLMs to analyze the incorrect solutions, providing insights into potential reasons for their failures, such as calculation errors, misunderstandings of the problem, or the application of incorrect formulas. The LLM also offers suggestions for enhancing the strategy, which may include adding subtasks, decomposing complex subtasks into simpler components, or revising ineffective subtasks. Subsequently, we utilize the LLM to refine the strategy based on this feedback.
>
> > Authors report the biggest gain from their method for symbolic reasoning tasks, but Addendum does not provide any examples of such tasks and the corresponding generated strategies.
>
> We would like to clarify that we have included a strategy and an execution example for the LLC task in Figure 27 on Page 30 of our paper. This example illustrates the application of our method on symbolic reasoning tasks.
>
> > How does strategy LLM generate multiple sets of strategies? Is it by asking it to re-run the same prompt multiple times or does it involve changing the original prompt?
>
> StrategyLLM generates multiple strategies by re-running the same prompt multiple times, utilizing temperature sampling to introduce variability in the outputs and allowing for the exploration of diverse strategies.
>
> > StrategyLLM might be susceptible to bias by generating discriminating strategies that may perform well on the training set, which is not addressed in the paper.
>
> We emphasize that StrategyLLM is specifically designed to address potential biases present in few-shot prompts and to enhance the generalizability of the prompts through the use of task-level strategies. Our comprehensive results demonstrate that the strategy-based prompts exhibit greater generalizability compared to traditional CoT prompts.

---

> > ### Comment · Reviewer_D2vF · 2024-08-11
> >
> > I'd like to thank the authors for their detailed explanations. After reading the concrete math example, I'm still unclear why missing the fact that we need the math books at both ends of the stack is something specific to CoT. This seems to be a miss on the LLM that is in charge of interpreting the question. It could be argued that if the strategy generator LLM misses this fact the same way that CoT's LLM missed it, then the proposed method by authors would make the same mistake as CoT.

---

> > > ### Author Response · Authors · 2024-08-11
> > > **Response to Reviewer D2vF**
> > >
> > > Thank you for your continued feedback and for highlighting the need for further clarification regarding the differentiation between the CoT approach and our proposed StrategyLLM method. We appreciate the opportunity to elaborate on this critical aspect.
> > >
> > > **Clarification:**
> > >
> > > We would like to clarify that the underlying LLM for the strategy generator is the same as the LLM being tested. Additionally, CoT and StrategyLLM leverage the same LLM for problem interpretation and resolution. However, the fundamental difference between these methods lies in their approach to guiding the LLM’s reasoning process.
> > >
> > > **Understanding LLM Capabilities:**
> > >
> > > It’s important to recognize that the LLM itself has the capacity to understand the details of a problem, such as the requirement to place math books at both ends of a stack. However, how effectively the LLM applies this understanding can vary significantly depending on the prompting method used.
> > >
> > > **The Role of Structured Strategy in Mitigating Misinterpretation:**
> > >
> > > The StrategyLLM approach is designed to mitigate the likelihood of misinterpretations through a structured reasoning process. Unlike CoT, which primarily guides the model to think step-by-step, StrategyLLM encourages a more holistic understanding of the problem by enforcing a structured strategy that must be followed consistently across examples.
> > >
> > > For instance, in the provided example, StrategyLLM ensures a systematic approach to solving it:
> > >
> > > *Step 1: Problem Understanding:* The strategy mandates a clear understanding of the problem requirements, including the necessity for math books at both ends of the stack.
> > >
> > > *Step 2: Identifying Relevant Concepts:* The strategy requires identifying the relevant counting or probability concepts needed to solve the problem, ensuring that the LLM considers the correct mathematical approach.
> > >
> > > *Step 3: Breakdown of the Problem:* The strategy explicitly requires identifying the critical elements of the problem, such as the placement of the math books and the arrangement of the remaining books.
> > >
> > > *Step 4: Calculation:* The strategy then guides the model to apply relevant mathematical concepts (e.g., permutations) to derive the correct solution.
> > >
> > > We can observe that the strategy itself does not directly identify critical details; rather, it strongly encourages the LLM to recognize these details for each problem. This approach significantly reduces the risk of overlooking crucial aspects.
> > >
> > > **Differentiating from CoT:**
> > >
> > > CoT allows for a more flexible reasoning process, which can sometimes lead to inconsistencies or omissions in understanding and solving problems. In the case of the example provided, CoT failed to capture the requirement of placing math books at both ends because the step-by-step reasoning did not enforce a structured breakdown of the problem. Instead, it focused on treating the two math books as a single entity, leading to an incorrect interpretation.
> > >
> > > **Conclusion:**
> > >
> > > In conclusion, while both CoT and StrategyLLM rely on the LLM’s interpretation, StrategyLLM’s structured approach reduces the likelihood of critical misinterpretations. This structured reasoning process ensures that all essential aspects of a problem are considered, leading to more accurate and reliable solutions.
> > >
> > > We hope this clarifies the distinction and addresses your concern. We appreciate your feedback and are committed to addressing any further questions or concerns you may have.

---

> > > > ### Comment · Reviewer_D2vF · 2024-08-11
> > > >
> > > > Thanks again for your thorough explanations. Can authors let me know what they expect to get from "gpt-3.5-turbo-16k-0613" without using any prompt engineering, few shot learning, and CoT if they pose the following question directly to the prompt?
> > > > "Suppose that I have 6 different books, 2 of which are math books. In how many ways can I stack my 6 books on a shelf if I want a math book on both ends of the stack?" My observation is that for this particular example, even a simple prompt without using more advanced approaches such as CoT or StrategyML should be able to solve the problem. Therefore, I suspect something in the CoT implementation is not working as intended and attributing a bug/issue in CoT implementation to its inferior performance compared with StrategyML is not advisable.

---

> > > > > ### Author Response · Authors · 2024-08-12
> > > > > **Response to Reviewer D2vF (Part 1)**
> > > > >
> > > > > Thank you for your feedback. We appreciate the opportunity for further clarification.
> > > > >
> > > > > **Clarification:**
> > > > >
> > > > > We would like to clarify that the example provided in our discussion was primarily intended to illustrate the motivation and benefits of our proposed StrategyLLM framework. The true measure of a method's effectiveness comes from its performance across diverse test sets of established benchmarks. Our extensive experimental results clearly demonstrate the superiority of StrategyLLM over baseline approaches, including CoT, across a wide range of LLMs and tasks.
> > > > >
> > > > > **Understanding LLM Behavior:**
> > > > >
> > > > > It’s important to recognize that LLMs, such as gpt-3.5-turbo, are autoregressive text generators. This means that the same problem can yield different solutions depending on the prompt used, or even when using the same prompt with temperature sampling. Therefore, it is not surprising that for certain examples, an LLM may generate a correct solution without any prompt engineering or advanced techniques. However, this does not imply a flaw in the CoT implementation. **We want to reassure you that there is no bug or issue in our CoT implementation.** While CoT does not guarantee outperformance over a simple zero-shot baseline on every individual example, its effectiveness is evident when evaluated across a larger test set, where it generally achieves higher accuracy.
> > > > >
> > > > > Our StrategyLLM framework is designed to increase the likelihood of obtaining correct answers by guiding the LLM through a structured reasoning process, ensuring a more consistent and reliable approach across different problems.
> > > > >
> > > > >
> > > > > **Prediction of Simple Prompt:**
> > > > >
> > > > > Given the optimization of frontier chat LLMs like “gpt-3.5-turbo” for step-by-step reasoning, we expect that even a simple prompt can yield a step-by-step solution for the provided example using “gpt-3.5-turbo-16k-0613”. However, predicting the exact reasoning steps in advance is quite challenging due to the inherent variability in LLM outputs. Since "gpt-3.5-turbo-16k-0613" has been deprecated, we used an improved version, "gpt-3.5-turbo-1106," to generate a prediction with a temperature setting of 0. The model produced the following solution:
> > > > >
> > > > > ```
> > > > > First, we need to consider the number of ways to arrange the 4 non-math books in the middle of the stack. This can be calculated using the formula for permutations of n objects, which is n!. In this case, there are 4 non-math books, so the number of ways to arrange them is 4!.
> > > > >
> > > > > Next, we need to consider the number of ways to arrange the 2 math books on the ends of the stack. Since there are 2 math books, the number of ways to arrange them is 2!.
> > > > >
> > > > > Finally, we can multiply these two numbers together to find the total number of ways to stack the books with a math book on both ends:
> > > > >
> > > > > 4! * 2! = 24 * 2 = 48
> > > > >
> > > > > So, there are 48 different ways to stack the 6 books on the shelf with a math book on both ends.
> > > > > ```
> > > > >
> > > > > This solution indeed derives the correct answer. However, as mentioned earlier, solving a single example correctly is not sufficient to establish the overall effectiveness of a method. To provide a broader comparison, we tested both the zero-shot baseline and the CoT approach on the CP dataset using "gpt-3.5-turbo-1106". The results were 33.0 for the zero-shot baseline and 36.5 for CoT, indicating that the CoT approach outperforms the simple baseline on the full test set.
> > > > >
> > > > > **A More Complicated Example:**
> > > > >
> > > > > The discussion regarding this complicated example is shown in the second part of our response (i.e., the next official comment).

---

> > > > > > ### Author Response · Authors · 2024-08-12
> > > > > > **Response to Reviewer D2vF (Part 2)**
> > > > > >
> > > > > > **A More Complicated Example:**
> > > > > >
> > > > > > We present a more complicated example, with a difficulty level of 5, to highlight the benefits of StrategyLLM. “ZS” refers to the zero-shot baseline without any prompt engineering.
> > > > > >
> > > > > > **Question:** How many ways are there to arrange $6$ beads of distinct colors in a $2 \times 3$ grid if reflections and rotations are considered the same? (In other words, two arrangements are considered the same if I can rotate and/or reflect one arrangement to get the other.)
> > > > > >
> > > > > > **Gold Answer:** 180
> > > > > >
> > > > > > **ZS Prediction:**
> > > > > >
> > > > > > ```
> > > > > > There are $6!$ ways to arrange the $6$ beads in a line. However, when considering a $2 \times 3$ grid, there are $6$ different ways to arrange the beads in a line that result in the same grid when considering reflections and rotations. Therefore, the total number of distinct arrangements is $\frac{6!}{6} = 120$.
> > > > > > ```
> > > > > >
> > > > > > **CoT Prediction:**
> > > > > >
> > > > > > ```
> > > > > > We can consider the $2 \times 3$ grid as a $6$-element cycle. There are $6!$ ways to arrange the $6$ beads in a line, but we must divide by $6$ to account for rotations. However, this overcounts the number of arrangements by a factor of $2$ because we are also counting reflections. Therefore, the final answer is $\frac{6!}{6 \cdot 2} = \boxed{60}$.
> > > > > > Answer: The answer is 60
> > > > > > ```
> > > > > > **StrategyLLM Prediction:**
> > > > > >
> > > > > > ```
> > > > > > 1. Understand the problem statement:
> > > > > >   - We have 6 beads of distinct colors.
> > > > > >   - We need to arrange these beads in a 2x3 grid.
> > > > > >   - Reflections and rotations are considered the same, meaning two arrangements are considered the same if one can be obtained from the other by rotating or reflecting.
> > > > > >
> > > > > > 2. Identify the counting or probability concept:
> > > > > >   - We need to use the concept of combinations to solve this problem.
> > > > > >
> > > > > > 3. Break down the problem:
> > > > > >   - We have 6 beads to arrange in a 2x3 grid.
> > > > > >   - We need to consider the symmetries of the grid to eliminate redundant cases.
> > > > > >   - We need to determine the number of distinct arrangements considering reflections and rotations.
> > > > > >
> > > > > > 4. Calculate the required counting or probability:
> > > > > >   - To calculate the number of distinct arrangements, we need to count the number of combinations that satisfy the given conditions.
> > > > > >   - We can use combinations to calculate this.
> > > > > >
> > > > > >   - Let's consider the possible arrangements:
> > > > > >     - We can choose any of the 6 beads for the first position in the grid.
> > > > > >     - We can choose any of the remaining 5 beads for the second position.
> > > > > >     - We can choose any of the remaining 4 beads for the third position.
> > > > > >     - We can choose any of the remaining 3 beads for the fourth position.
> > > > > >     - We can choose any of the remaining 2 beads for the fifth position.
> > > > > >     - We can choose the last remaining bead for the sixth position.
> > > > > >
> > > > > >   - Total number of distinct arrangements = 6 x 5 x 4 x 3 x 2 x 1 = 720.
> > > > > >
> > > > > > 5. Simplify or reduce the problem:
> > > > > >   - We need to consider the symmetries of the grid to eliminate redundant cases.
> > > > > >   - The 2x3 grid has rotational symmetry of order 2, meaning it can be rotated 180 degrees to look the same.
> > > > > >   - The 2x3 grid also has reflectional symmetry, meaning it can be reflected horizontally or vertically to look the same.
> > > > > >   - Therefore, we need to divide the total number of distinct arrangements by 2 to account for the rotational symmetry.
> > > > > >   - We also need to divide the total number of distinct arrangements by 2 to account for the reflectional symmetry.
> > > > > >
> > > > > > 6. Calculate the total number of distinct arrangements:
> > > > > >   - Total number of distinct arrangements = 720 / 2 / 2 = 180.
> > > > > >
> > > > > > Answer: The number of ways to arrange 6 beads of distinct colors in a 2x3 grid, considering reflections and rotations as the same, is 180
> > > > > > ```
> > > > > >
> > > > > > We observe that both ZS and CoT handle rotations and reflections incorrectly and fail to provide adequate details on how to address them. In contrast, StrategyLLM offers comprehensive details and applies correct logic, resulting in the correct solution. This example clearly demonstrates the superiority of StrategyLLM in complex reasoning tasks.
> > > > > >
> > > > > > **Conclusion:**
> > > > > >
> > > > > > In summary, while a simple prompt may occasionally yield correct solutions, our findings indicate that advanced methods like CoT, and more importantly, our StrategyLLM framework, provide more reliable and consistent results across a wide range of examples. We hope this explanation addresses your concerns and clarifies the rationale behind our approach.

---

> > > > > > > ### Comment · Area_Chair_PNDq · 2024-08-13
> > > > > > >
> > > > > > > Dear Reviewer,
> > > > > > >
> > > > > > > I would appreciate if you could comment on the author's rebuttal, in light of the upcoming deadline.  Did the interaction help to iron out any remaining criticism or doubt?
> > > > > > >
> > > > > > > Thank you,
> > > > > > > Your AC

---

> > > > > > > > ### Comment · Reviewer_D2vF · 2024-08-14
> > > > > > > >
> > > > > > > > Thanks again for the detailed explanations and the new example shared by the authors. In light of the new comments, I changed my score from borderline reject to borderline accept. If more complex such as the one shared in your latest comments are the difference maker between zs vs. cot vs. strategyLLM, I encourage you to spell out the complexities in such questions in your paper and explain why strategyLLM can do a better job compared with other approaches like cot.

---

> > > > > > > > > ### Author Response · Authors · 2024-08-14
> > > > > > > > > **Response to Reviewer D2vF**
> > > > > > > > >
> > > > > > > > > Thank you for your thoughtful consideration and for adjusting your evaluation based on our latest discussions. We truly appreciate your willingness to engage deeply with our work. We fully understand the importance of clearly articulating how StrategyLLM’s structured reasoning framework helps to mitigate common pitfalls that may occur in CoT or ZS approaches. Based on your valuable feedback, we will revise our paper to better highlight these complexities and provide detailed explanations of why StrategyLLM excels in handling them.

---

> > > > > > > > > > ### Comment · Reviewer_D2vF · 2024-08-14
> > > > > > > > > >
> > > > > > > > > > Thanks very much again for your responses and clarifications and willingness to incorporate the discussed changes in the revised version of your work!

---

### Official Review · Reviewer_s895 · 2024-07-10

**Soundness:** 4
**Presentation:** 4
**Contribution:** 3
**Rating:** 6
**Confidence:** 4

**Summary:**

This work uses an LLM to describe the strategy that it should use to solve classes of problems, and then applies that strategy to the different test cases as required.  In addition, there is a caching and evaluation process whereby these strategies are improved during in an in-context learning prompt training/refinement cycle.  This approach is tested across a number of datasets and underlying (commercial and open-source) LLMs.

**Strengths:**

Well written paper, with a clear idea and thorough experimentation.

Good to see both commercial and open-source models being tested side-by-side.

The strategy artifacts produced by the model are very interesting in their own right.  They are produced by the 'portable' prompts in the Appendix, however by the time they have been validated/cached/refined, they will likely be model-specific.

**Weaknesses:**

The actual prompt (Figure 7, in Appendix C) on which *the whole approach relies* should be included in the body of the paper.

Text Tweak:

* L560: "the optimal strategies for various datasets" - seems unlikely.  Perhaps "the best strategies discovered ..." ?

**Questions:**

* L221-233: Clarification: Is the Strategy Generator the same LLM as being 'tested' - or is the strategy being generated originally by GPT-3.5?

**Limitations:**

As described in the text, this approach is very dependent on whether the underlying LLM has a 'high level' understanding of how to approach the problem domain.  This is a dependency that somewhat undercuts the broad claims of the work.

---

> ### Author Rebuttal · Authors · 2024-08-07
>
> Thank you for your insightful comments. We address each question as follows:
>
> > Regarding the actual prompt and text tweak
>
> We appreciate your suggestions. We will incorporate these revisions in the new version of our paper.
>
> > Is the Strategy Generator the same LLM as being 'tested' - or is the strategy being generated originally by GPT-3.5?
>
> We would like to clarify that the underlying LLM for the strategy generator, executor, optimizer, and evaluator is the same as the LLM being tested.

---

> > ### Comment · Reviewer_s895 · 2024-08-11
> >
> > Thank you for your responses to my questions.  Having read those (and those in response to the other reviewers' questions), I'll stick with my original Rating (with apologies to the AC for not being making the final decision any easier)

---

### Official Review · Reviewer_WbXZ · 2024-07-12

**Soundness:** 3
**Presentation:** 3
**Contribution:** 3
**Rating:** 6
**Confidence:** 3

**Summary:**

This paper introduces a novel prompting strategy for large language models called StrategyLLM, which it employs to solve a range of tasks in math, commonsense reasoning, word sorting, and last letter concatenation. The approach involves (1) prompting a language model to generate task-specific instance-general strategies (sequences of subtasks) from a few examples of a given task, (2) prompting a language model to execute these strategies in order to grade them, (3) prompting a language model to refine strategies, (4) evaluating the strategies on a validation set to select inference time strategies, and (5) employing one or more of the selected strategies to solve new instances of the tasks of interest. The key attribute of this approach is that the strategies generated are not instance-specific, and indeed the paper finds improved generalization compared with prompting with instance-specific solutions as in the CoT baseline. The paper evaluates the StrategyLLM approach against a selection of standard prompting baselines, finding improvements across all tasks considered and across all models considered, and provides additional analyses.

**Strengths:**

The main high-level thrust of the paper, that LLM can generate strategies specific to a task, and then carry out these strategies is an important direction. This paper shows that LLMs can indeed use strategy generation to achieve greater generalization than an LLM performing in-context learning from example solutions. The experiments are adequately wide-ranging and the results are compelling; strategy generation as employed in StrategyLLM yields improvements over the baselines compared against. These baselines are well selected, and while they don't represent the full suite of prompting strategies that people employ with LLMs today, they make for a fair comparison that allows for measuring the effect of the core idea of StrategyLLM, having the LLM generate task-specific strategies.

Prompting methods to elicit improved reasoning from large language models is an active area, ripe with a large number of unexplored ideas. This paper does a good job of placing its approach in the context of some of the recent work in this area. This is not the first paper to employ prompting strategies that share this high-level thrust, and as the paper cites a closely related learning-to-plan approach [15] has previously been explored.

Overall the paper is clearly written, though in the Weaknesses section I remark on how the SolutionLLM baseline should be explained more clearly.

**Weaknesses:**

The approach in StrategyLLM is modestly complicated. For example, a simpler approach that still tests the core idea of the paper would be to keep the strategy generator and inference steps, but skip the intermediate grading and optimization of the strategies. Would the method work without these extra prompting phases, or are they essential to the method? An ablation would be valuable. I would expect these phases are essential, since they provide a data-driven validation of strategies before they are employed. This raises another natural question which is: is it the data-driven validation of the prompt (in this case, a strategy) that is the main advantage of StrategyLLM, or is it specifically that the strategies are not instance-specific? An experiment that compares with a prompt selection approach analogous to StrategyLLM's (i.e. generating multiple prompts, grading them on a validation set, refining them as needed, using a validation set to select between them, and finally employing SC or ZC to get a final answer), but  generating e.g. few-shot prompts instead of strategies would yield a fairer comparison.

In my view, the SolutionLLM baseline is not explained clearly. Two points in particular could use clarification. First, state more clearly what is meant by a "solution"; this is understandable by looking at the prompt in the appendix, but could use clarification in the text. Second, state explicitly how the SolutionLLM prompt is used; the prompt in the appendix merely outputs the solution to a specific example (I presume this is a dev/validation example), but it is not stated (if I am not mistaken) how this results in answers for novel tasks.

**Questions:**

Clarity: Line 85 describes the role of the evaluator, constructing few-shot prompts for a strategy "using itself and its execution result". Please clarify the interface of the evaluator (line 122), and specifically what the form of the few shot examples p_{st} are. Providing the prompt used by the evaluator would be a welcome change, as it is the main absent component in the current revision.

See also the clarify questions about the SolutionLLM baseline in the Weaknesses section of the review.

Let's have some discussion on how two rounds of validation are used before applying the strategy to the task instances of interest. Specifically, the execution phase grades a strategy on a small set of task examples $\mathcal{E}$ (the same examples used to generate the strategy), and then the evaluator E grades the best performing strategies on a larger validation set $\mathcal{V}$. Both rounds of validation are being used for filtering the set of strategies. In the paper, these are presented as separate phases of the StrategyLLM method with different type signatures and different descriptions. Clarity-wise, I fear this might be obscure the fact they represent two rounds of validation and filtering.

The final prompts used during inference include both the strategy generated as well as consistent few-shot prompts (consistent because they all employ the same strategy). Do you have evaluation of relative value of the presence of the strategy vs the presence of the consistent few-shot prompts.

I would encourage you to write out what StrategyLLM is in contrast with StrategyLLM-SC and StrategyLLM-ZS; my assumption is that StrategyLLM employs only one selected strategy, but I don't see this stated explicitly.

Terminology: Why do you describe your approach as involving multiple LLM agents? To me it seems like it is simply an algorithm that calls a single LLM with multiple prompts orchestrated by an (ordinary) algorithm. I understand that there are separate components for G, X, O, E, and inference, and my question is specifically around why you describe each of G, X, O, and E as a separate "agent".

**Limitations:**

The paper includes an adequate Limitation and Impact section, though it does not address societal impacts in particular. Per the checklist response, this is because the paper aims "to enhance the capabilities of large language models in common task-solving scenarios and does not introduce privacy, security, or fairness issues."

---

> ### Author Rebuttal · Authors · 2024-08-07
>
> Thank you for your insightful comments. We address each question as follows:
>
> > A simpler approach that still tests the core idea of the paper would be to keep the strategy generator and inference steps, but skip the intermediate grading and optimization of the strategies. Would the method work without these extra prompting phases, or are they essential to the method?
>
> We would like to emphasize that these phases are indeed essential to the method. Specifically, StrategyLLM relies on the strategy executor to apply the generated strategies to the provided examples. The execution results are necessary for formulating the strategy-based prompts. Additionally, the strategy evaluator plays a vital role in determining the most effective strategy for inference. Without these phases, we would be unable to generate the few-shot prompts necessary for each strategy or to make informed decisions regarding which strategy to employ.
>
> > Is it the data-driven validation of the prompt (in this case, a strategy) that is the main advantage of StrategyLLM, or is it specifically that the strategies are not instance-specific? An experiment that compares with a prompt selection approach analogous to StrategyLLM's (i.e. generating multiple prompts, grading them on a validation set, refining them as needed, using a validation set to select between them, and finally employing SC or ZC to get a final answer), but generating e.g. few-shot prompts instead of strategies would yield a fairer comparison.
>
> We appreciate your valuable suggestion. In response, we implement a baseline approach, which involves initially generating $n$ prompts, retaining $k$ of these for validation, and subsequently selecting the prompt with the highest validation accuracy for inference. The results of Meta-Llama-3-8B-Instruct on the CP, StrategyQA, and MA datasets are presented in the table below. We can observe that StrategyLLM significantly outperforms this baseline across all datasets, highlighting the benefits of integrating generalizable strategies.
>
> |     | CP   | StrategyQA | MA   | Avg   |
> |------|------|-----|-----|------|
> | Baseline    | 19.5 | 71.0 | 53.3 | 47.9 |
> | StrategyLLM | 24.5 | 74.0 | 64.7 | **54.4** |
>
> > Regarding the SolutionLLM baseline
>
> A solution is a step-by-step reasoning path. In this baseline, we utilize LLMs to generate a solution for each example in the prompt, which is the same as the examples used in the CoT prompt. These examples, along with their respective solutions, are then combined to formulate a few-shot prompt for inference, analogous to the CoT approach.
>
> > Regarding the evaluator and constructing the few-shot prompt
>
> Unlike other agents, the strategy evaluator does not utilize a specific prompt. Instead, to assess the effectiveness of a strategy, the evaluator constructs a few-shot strategy-based prompt tailored to that particular strategy. An illustrative example of the strategy-based prompt is depicted in Figure 3 of our paper. This prompt is structured as follows: **Strategy + (Question_1, Solution_1, Answer_1) + … + (Question_n, Solution_n, Answer_n)**. The solutions, Solution_1 through Solution_n, are derived during the strategy execution phase by applying the strategy to the $n$ examples in the prompt. The strategy evaluator then infers using each prompt on the validation set and calculates its validation accuracy.
>
>
> >  In the paper, the execution and validation are presented as separate phases of the StrategyLLM method with different type signatures and different descriptions. Clarity-wise, I fear this might be obscure the fact they represent two rounds of validation and filtering.
>
> Thank you for your observation. We acknowledge that these phases represent two rounds of validation. However, they differ in their approach to obtaining solutions. During the execution phase, we directly prompt the LLM to generate a solution for a specific example by following the strategy. Next, in the validation phase, the strategy and the examples with their solutions are utilized to construct the strategy-based prompt. LLMs use this strategy-based prompt to infer solutions for each validation example. We will clarify these distinctions in the revised version of our paper to enhance understanding.
>
> > Do you have evaluation of relative value of the presence of the strategy vs the presence of the consistent few-shot prompts.
>
> We conduct experiments to assess inference performance without the presence of the strategy, utilizing only the consistent few-shot prompt. The results for Meta-Llama-3-8B-Instruct are summarized in the table below. The findings indicate that omitting the strategy leads to a decrease in performance. This is because the consistent prompt merely illustrates how to apply the strategy to specific examples and may not encapsulate all the nuances of the strategy itself.
>
> |    | CP | StrategyQA | MA | Avg |
> |-----|------|------|------|-------|
> | StrategyLLM wo/ strategy | 24.5 | 66.0  | 63.3 | 51.3 |
> | StrategyLLM   | 24.5 | 74.0  | 64.7 | **54.4**  |
>
> > I would encourage you to write out what StrategyLLM is in contrast with StrategyLLM-SC and StrategyLLM-ZS; my assumption is that StrategyLLM employs only one selected strategy, but I don't see this stated explicitly.
>
> Yes. StrategyLLM employs the most effective strategy identified during our experiments. We will state this clearly in the revised version of our paper.
>
> > Why do you describe your approach as involving multiple LLM agents?
>
> We follow the terminology used in existing popular agent frameworks such as BabyAGI  (https://github.com/yoheinakajima/babyagi). Multiple agents collaborate to accomplish a task, with each agent assigned a subtask of the task. While the underlying LLMs for these agents can differ, our experiments utilize the same LLM across all agents to ensure a fair comparison.

---

> > ### Comment · Reviewer_WbXZ · 2024-08-13
> >
> > Thank you for your responses to my questions and for the new experimental results.
> >
> > The results are helpful for clarifying the necessity and relative importance of the different facets of the StrategyLLM approach.
> >
> > For the parts of your response that clarify ambiguities in the paper, I encourage you to incorporate those clarifications into the paper itself.

---

> > > ### Author Response · Authors · 2024-08-13
> > > **Response to Reviewer WbXZ**
> > >
> > > Thank you for your thoughtful feedback and for taking the time to review our rebuttal. We are pleased to hear that the new results have helped clarify the necessity and importance of the various facets of our StrategyLLM approach. We appreciate your suggestions and will incorporate these clarifications into our revised manuscript to enhance the overall clarity and comprehensiveness of our work.

---

### Official Review · Reviewer_CCvY · 2024-07-18

**Soundness:** 3
**Presentation:** 3
**Contribution:** 3
**Rating:** 6
**Confidence:** 5

**Summary:**

This paper proposes StrategyLLM, a pipeline for improving the few-shot reasoning performance. The main intuition is that when solutions to few-shot exemplars are inconsistent in terms of the reasoning process, the performance can be suboptimal compared to those with consistent solutions. Based on this intuition, StrategyLLM includes 3 prompting-based components: strategy generator, executor, and optimizer. The strategy generator generates task-level reasoning guidelines, the executor generates solutions to few-shot exemplars based on the generated strategies, and the optimizer improves the strategy when the accuracy on examples is lower than a threshold. Experiments on several reasoning benchmarks demonstrate that StrategyLLM outperforms other prompting methods, including CoT and plan-and-solve.

**Strengths:**

1. The empirical results are good compared to the baselines prompting methods, and the evaluation covers multiple LLMs.

2 There is a thorough breakdown analysis on different number of samples, different number of optimization steps, gap to the oracle selection accuracy, etc.

**Weaknesses:**

1. The few-shot setting for StrategyLLM is unclear. Specifically, the paper says that a strategy is applied when it passes an accuracy threshold, which is <1. Does it mean that when using the few-shot prompt constructed by StrategyLLM, some examples have wrong solutions? If this is the case, it is surprising that StrategyLLM still outperforms the baselines.

2. Have the authors tried self-consistency with a single strategy? How much performance gain is from aggregating multiple strategies compared to sampling from a single strategy?

3. On MATH, it is unnatural to construct separate prompts per category. Have the authors tried constructing a single prompt for all MATH problems, and how does it compare to baselines?

**Questions:**

1. Please clarify the few-shot setting for StrategyLLM. Specifically, does the few-shot prompt constructed by StrategyLLM contain wrong solutions when the generated strategies do not pass all examples?

2. Have the authors tried self-consistency with a single strategy? How much performance gain is from aggregating multiple strategies compared to sampling from a single strategy?

3. Have the authors tried constructing a single prompt for all MATH problems, and how does it compare to baselines?

**Limitations:**

Yes.

---

> ### Author Rebuttal · Authors · 2024-08-07
>
> Thank you for your insightful comments. We address each question as follows:
>
> > The paper says that a strategy is applied when it passes an accuracy threshold, which is <1. Does it mean that when using the few-shot prompt constructed by StrategyLLM, some examples have wrong solutions?
>
> Yes, we would like to clarify the following points: (1) In most scenarios, we set the accuracy threshold at 0.75, and the number of examples used is fewer than 8, there can be at most 1 example with an incorrect solution in these scenarios. (2) These incorrect examples may not largely degrade the performance. As pointed out by existing research (e.g., [1]),  the relevance of the solution to the question and the presentation of intermediate reasoning steps can be more critical than the correctness of the final answer.
>
> [1] Towards Understanding Chain-of-Thought Prompting: An Empirical Study of What Matters. ACL 2023. https://aclanthology.org/2023.acl-long.153/
>
> > Have the authors tried self-consistency with a single strategy? How much performance gain is from aggregating multiple strategies compared to sampling from a single strategy?
>
> We have explored self-consistency using a single strategy. The performance of Meta-Llama-3-8B-Instruct on the CP, StrategyQA, and MA datasets is summarized in the table below. "StrategyLLM-Single-SC" refers to self-consistency applied with the best-performing strategy identified, which generates a set of solutions via
> temperature sampling to obtain multiple answers. The results indicate that StrategyLLM-Single-SC underperforms compared to StrategyLLM-SC and even StrategyLLM, highlighting the advantages of aggregating multiple strategies.
>
> |                    | CP   | StrategyQA | MA   | Avg   |
> |---------------------------|------|------------|------|-------|
> | StrategyLLM               | 24.5 | 74.0       | 64.7 | 54.4  |
> | StrategyLLM-Single-SC | 23.5 | 69.8       | 59.3 | 50.9  |
> | StrategyLLM-SC | 25.0 | 74.0       | 66.0 | **55.0**  |
>
> > Have the authors tried constructing a single prompt for all MATH problems, and how does it compare to baselines?
>
> Thank you for your suggestion. We would like to clarify that constructing a distinct prompt for each subject may be more appropriate as the strategy can be more specific to the subject and incorporate more domain-specific knowledge of it.
>
> Additionally, to address your question, we conduct experiments using a single CoT prompt and a strategy-based prompt, both utilizing the same 7 examples, with each example drawn from a different subject in the MATH benchmark. Both the StrategyLLM and CoT prompts were evaluated on the test sets across all MATH subjects. The results for Meta-Llama-3-8B-Instruct are presented in the table below. The findings indicate that StrategyLLM consistently outperforms CoT across all subjects, demonstrating the advantages of our framework.
>
> |                    | AL   | PA | IA | CP | NT | GE | PC | Avg |
> |---------------------------|------|------------|------|-------|------|-------|------|------|
> | CoT | 36.5 | 44.5 | 4.5 | 23.0 | 18.0 | 16.0 | 13.5 | 22.3 |
> | StrategyLLM               | 39.0 | 48.5 | 8.5 | 24.0 | 22.5 | 18.0 | 17.5 | **25.4** |

---

> > ### Comment · Reviewer_CCvY · 2024-08-12
> >
> > I thank the authors for adding additional evaluation in the rebuttal. I will keep my original rating.

---

### Decision · Program_Chairs · 2024-09-25

**Decision:**

Accept (poster)

**Comment:**

After a fruitful discussion, the reviews generally agree that the paper is well written and that the core contribution is clear and insightful. They have also provided a positive evaluation of the experiments, which consider a variety of sensible models and baselines and yield compelling results.  This is a solid paper overall, and the majority of issues brought up by the reviewers have been solved during the rebuttal stage.